# Epstein-Barr virus latent membrane protein 1 subverts IMPDH pathways to drive B-cell oncometabolism

Eric M. Burton[1,2,3], Davide Maestri[1,2,3], Shaowen White[1,2,3], Jin-Hua Liang[1,2,3], Bidisha Mitra[1,2,3], John M. Asara[4], Benjamin E. Gewurz[1,2,3,5]*

1 Division of Infectious Diseases, Department of Medicine, Brigham and Women's Hospital, Boston, Massachusetts, United States of America, 2 Center for Integrated Solutions for Infectious Diseases, Broad Institute of Harvard and MIT, Cambridge, Massachusetts, United States of America, 3 Department of Microbiology, Harvard Medical School, Boston, Massachusetts, United States of America, 4 Division of Signal Transduction, Beth Israel Deaconess Medical Center and Department of Medicine, Harvard Medical School, Boston, Massachusetts, United States of America, 5 Harvard Program in Virology, Harvard Medical School, Boston, Massachusetts, United States of America

* bgewurz@bwh.harvard.edu

## Abstract

Epstein-Barr virus (EBV) is associated with multiple types of cancers, many of which express the viral oncoprotein Latent Membrane Protein 1 (LMP1). LMP1 contributes to both epithelial and B-cell transformation. Although metabolism reprogramming is a cancer hallmark, much remains to be learned about how LMP1 alters lymphocyte oncometabolism. To gain insights into key B-cell metabolic pathways subverted by LMP1, we performed systematic metabolomic analyses on B cells with conditional LMP1 expression. This approach highlighted that LMP highly induces *de novo* purine biosynthesis, with xanthosine-5-P (XMP) as one of the most highly LMP1-upregulated metabolites. Consequently, IMPDH inhibition by mycophenolic acid (MPA) triggered death of LMP1-expressing EBV-transformed lymphoblastoid cell lines (LCL), a key model for EBV-driven immunoblastic lymphomas. Whereas MPA instead caused growth arrest of Burkitt lymphoma cells with the EBV latency I program, conditional LMP1 expression triggered their death, and this phenotype was rescuable by gua-nosine triphosphate (GTP) supplementation, implicating LMP1 as a key driver of B-cell GTP biosynthesis. Although both IMPDH isozymes are expressed in LCLs, only IMPDH2 was critical for LCL survival, whereas both contributed to proliferation of Burkitt cells with the EBV latency I program. Both LMP1 C-terminal cytoplasmic tail domains critical for primary human B-cell transformation were important for XMP pro-duction, and each contributed to LMP1-driven Burkitt cell sensitivity to MPA. Metab-olomic analyses further highlighted roles of NF-kB, mitogen activated kinase, and protein kinase C downstream of LMP1 in support of XMP abundance. Of these, only protein kinase C activity was important for supporting GTP levels in LMP1 express-ing Burkitt cells. MPA also de-repressed EBV lytic antigens, including LMP1 itself in

**Data availability statement:** All relevant data are in the manuscript and its Supporting information files.

**Funding:** This study was supported by the American Cancer Society, PF-23-898493-01-TBE to EMB, PF-24-1308318-01-TBE to SW, the Division of Intramural Research, National Institute of Allergy and Infectious Diseases, T32AI007245 to EMB and T32AI007061 to SW, the Division of Cancer Prevention, National Cancer Institute, CA275301 to BEG, the Division of Cancer Epidemiology and Genetics, National Cancer Institute, CA228700 to BEG, the Division of Intramural Research, National Institute of Allergy and Infectious Diseases, AI164709 to BEG; the National Cancer Institute, CA275301 to BEG and the Division of Cancer Prevention, National Cancer Institute, CA269043 to BEG. The funders had no role in study design, data collection and analysis, decision to publish, or preparation of the manuscript.

**Competing interests:** The authors have declared that no competing interests exist.

latency I Burkitt cells, highlighting crosstalk between the purine biosynthesis pathway and the EBV epigenome. These results suggest novel oncometabolism-based therapeutic approaches to LMP1-driven lymphomas.

## Author summary

Altered metabolism is a hallmark of cancer, yet much remains to be learned about how EBV rewires host cell metabolism to support multiple malignancies. While LMP1 is a well-described oncogene, knowledge has remained incomplete about how LMP1 alters host cell oncometabolism to aberrantly drive infected B-cell growth and survival. Likewise, it has remained unknown whether LMP1 expression creates metabolic vulnerabilities that can be targeted by small molecule approaches to trigger EBV-transformed B-cell programmed cell death. We therefore used metabolomic profiling to define how LMP1 signaling remodels the B-cell metabolome. We found that LMP1 upregulated purine nucleotide biosynthesis, likely to meet increased demand. Consequently, LMP1 expression sensitized Burkitt B-cells to growth arrest upon inosine monophosphate dehydrogenase blockade. Thus, while LMP1 itself may not be a therapeutic target, its signaling induces dependence on downstream druggable host cell nucleotide metabolism enzymes, suggesting rational therapeutic approaches.

## Introduction

Epstein-Barr virus (EBV) persistently infects over 90% of adults worldwide. EBV causes infectious mononucleosis, is a key multiple sclerosis trigger and contributes to 200,000 cancers per year. EBV is associated with a wide range of lymphomas, including endemic Burkitt lymphoma (BL), Hodgkin lymphoma, post-transplant lymphoproliferative diseases (PTLD), T and NK cell lymphomas. EBV is also highly associated with nasopharyngeal carcinoma and gastric carcinoma [1–6].

To establish lifelong colonization of the memory B cell compartment, EBV uses a series of latency programs, in which combinations of oncogenic Epstein-Barr nuclear antigens (EBNA) and latent membrane proteins (LMP), as well as non-coding factors such as EBERS, BARTs, and BART-miRNAs, are expressed [7,8]. The highly B-cell transforming latency III program is comprised LMP1 and LMP2A, all six EBNA and non-coding RNAs [9–11]. LMP1 and LMP2A mimic signaling from activated CD40 and B-cell receptors, respectively. Latency III is hypothesized to drive infected B-cells into secondary lymphoid germinal centers, within which infected B-cells switch to the latency II program, comprised of EBNA1, LMP1 and LMP2A. Germinal center cytokines boost LMP1 expression through JAK/STAT signaling [12,13]. Upon memory B-cell differentiation, EBV switches to the latency I program, in which EBNA1 is the only EBV-encoded protein expressed, together with non-coding RNAs. Most Burkitt lymphomas express latency I [6,14–16].

LMP1 is capable of transforming B lymphocytes and epithelial cells [17–23]. LMP1 is comprised of a 24 residue N-terminal cytoplasmic tail, six transmembrane (TM) domains and a 200 residue C-terminal cytoplasmic tail [10,11,24,25]. LMP1 TM domains drive lipid raft association and constitutive signaling by cytoplasmic tail regions [26–28]. Two LMP1 C-terminal tail regions, termed C-terminal activating region (CTAR) or transformation effector site (TES), are essential for primary human B-cell transformation. CTAR1/TES1 activates canonical and non-canonical NF-κB, PI3 kinase, MAP kinase (MAPK), protein kinase C and JAK/STAT signaling, whereas CTAR2/TES2 activates canonical NF-κB, MAP, IRF7 and P62 pathways [10,11,25,26,29–36]. The LMP1 CTAR3 region activates JAK/STAT and SUMOylation pathways [37–39].

EBV oncogenes have not yet proven to be druggable *in vivo*, though induce downstream host cell dependencies that may instead be attractive therapeutic targets, including within remodeled oncometabolism pathways. EBV-driven metabolic remodeling is necessary for primary human B cell transformation into immortalized lymphoblastoid cell lines (LCL), a major PTLD model [40–43]. For instance, EBV manipulates nucleotide metabolism to support oncogene-driven demand [44–46], in which infected cells become highly dependent upon EBV upregulated nucleotide metabolism pathways, including one-carbon metabolism [42]. EBNA2 and LMP1 jointly induce the *de novo* cytidine biosynthesis pathway, which then exerts critical roles in EBV-transformed B-cell proliferation [40]. However, while metabolism reprogramming is a hallmark of cancer [47], LMP1 effects on host B-cell metabolism pathways remain to be fully characterized.

Here, we use metabolomic approaches to characterize LMP1 roles in host B-cell oncometabolism remodeling. Liquid chromatography/mass spectrometry (LC/MS) profiling identified that LMP1 reprogrammed a wide range of metabolic pathways, in particular purine metabolism. LMP1 upregulated levels of the guanosine monophosphate precursor xanthosine-5-phosphate (XMP), which is produced by activity of the enzymes inosine monophosphate dehydrogenase 1 and 2 (IMPDH1/2). IMPDH blockade by mycophenolic acid (MPA) or by CRISPR editing triggered LCL death, which was rescuable by guanosine triphosphate (GTP) supplementation. LMP1 sensitized latency I Burkitt cells to killing upon MPA treatment, and this phenotype was dependent upon TES1 and TES2 signaling. Multiple LMP1 activated pathways support XMP and GTP steady state levels in latency III. Within latency I, IMPDH blockade triggered growth arrest and reshaped the latency I EBV epigenome.

## Results

### Metabolomic analyses highlights key pathways targeted by LMP1

To gain insights into LMP1 B-cell oncometabolism remodeling roles, we constructed Burkitt lymphoma B-cell lines with conditional LMP1 expression. We validated that doxycycline-induced LMP1 expression triggered NF-κB pathway activation and LMP1 target gene expression in EBV+ Daudi EBV+ as well as in EBV-negative Akata Burkitt cells (S1A–S1D Fig). To confirm efficient LMP1 expression in Daudi and Akata cells, we utilized well-characterized plasma membrane markers upregulated by LMP1 signaling, Fas and ICAM-1. Fas was chosen for Daudi and ICAM-1 for Akata, due to their low basal levels in these Burkitt contexts [48]. We then performed polar metabolite LC/MS profiling of both Daudi and Akata cell models, mock induced or induced for LMP1 expression for 24 hours by doxycycline addition (Fig 1A). LMP1 significantly induced 124 and reduced 10 Daudi metabolites (Fig 1B and S1 Table). Similarly, LMP1 induced 56 and reduced 11 Akata metabolites (Fig 1C and S1 Table). Metabolite pathway impact analysis [49] identified that purine metabolism was amongst the most highly LMP1- induced metabolic pathway in both Burkitt model systems (Fig 1D and 1E).

The purine biosynthesis metabolite xanthosine-5-phosphate (XMP), produced from the precursor inosine monophosphate by the enzymes inosine monophosphate dehydrogenase (IMPDH) 1 and 2, was one of the most highly LMP1 upregulated metabolite common to both Akata and Daudi cells (Fig 1F and 1G, and S2 Table). XMP is converted into guanosine monophosphate (GMP), guanosine diphosphate (GDP) and guanosine triphosphate (GTP) (Fig 1G). However, LMP1 more modestly increased GMP, GDP and GTP levels, suggesting that LMP1 may also increase their consumption. Further underscoring LMP1 nucleotide metabolism remodeling, LMP1 increased steady state levels of the key purine

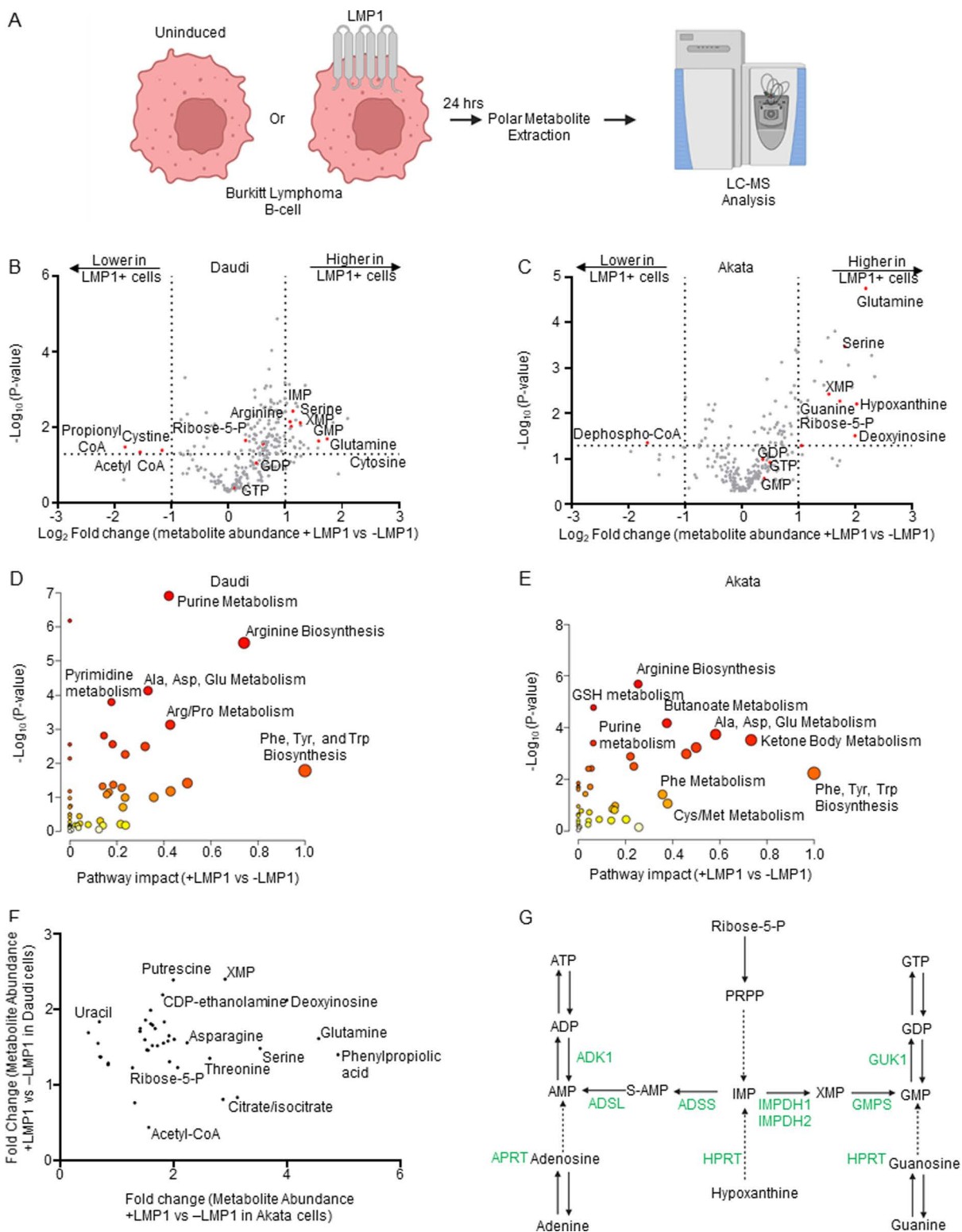

**Fig 1. LMP1-mediated B-cell metabolome remodeling.** (A) Metabolomic experiment workflow. Conditional LMP1 Daudi or Akata Burkitt cells were mock induced or induced by doxycycline (250ng/ml) for LMP1 expression for 24h. Polar metabolites were analyzed by targeted metabolomic analysis.

(B) Volcano plot analysis of liquid chromatography mass spectrometry (LC-MS) analysis of n = 3 replicates of Daudi cells mock induced or induced for LMP1 expression. Positive fold changes indicate higher metabolite concentrations in LMP1 + vs LMP1- cells. Selected host cell metabolites induced vs. suppressed by LMP1 are indicated. (C) Volcano plot analysis of LC-MS analysis of n = 6 replicates of Akata cells mock induced or induced for LMP1 expression. (D) Metaboanalyst analysis of LMP1 driven Daudi cell metabolic pathway impact from the data presented in (B), using data from significantly changed metabolites in LMP1 + vs LMP1- cells (p value > 0.05). Higher pathway impact values indicates stronger effects of conditional LMP1 expression on the indicated pathway. Purine metabolism was amongst the most highly LMP1 impacted pathways. (E) Metaboanalyst pathway impact analysis of LMP1 + vs LMP1- Akata cells from the data presented in (C). (F) Volcano plot analysis cross-comparing fold change of metabolite abundances in LMP1+ versus LMP1- Akata cells (x-axis) vs Daudi cells (Y-axis). Shown are metabolites whose abundances were LMP1 increased by ≥1.2 fold in both cell models. Selected metabolites are annotated, including xanthosine-5-phosphate, which was highly LMP1-induced under both conditions. (G) Purine metabolism pathways. The de novo pathway uses Ribose-5-phosphate and PRPP to generate inosine monophosphate (IMP), whereas the salvage pathway metabolizes hypoxanthine into IMP. IMP can be converted by IMPDH1/2 to xanthosine monophosphate (XMP) and subsequently to guanosine monophosphate (GMP). Alternatively, IMP can be converted to adenosine monophosphate (AMP). Created in BioRender. Burton, E. (2025) https://BioRender.com/x8l0ikn.

nucleotide biosynthetic building block ribose-5-phosphate (Fig 1F and 1G). LMP1 also increased levels of the hypoxanthine precursor deoxyinosine, suggesting that LMP1 may also support purine demand through upregulation of purine salvage metabolism (Fig 1F and 1G).

LMP1 also highly induced amino acid metabolism in both Daudi and Akata. Pathway impact analysis [20] highlighted that glutamine metabolism was highly induced in both Burkitt models (Fig 1D and 1E). Notably, glutamine plays major roles in de novo purine and pyrimidine synthesis and was the most highly LMP1 induced amino acid in Akata and Daudi (Fig 1F). LMP1 also significantly increased the abundance of serine, which is a major one carbon metabolism donor for de novo purine biosynthesis, including in newly EBV-infected and EBV-transformed LCLs [42]. Together, these metabolomic profiling results suggest that LMP1 remodels B-cell nucleotide and amino acid metabolism pathways to support nucleotide demand.

## Latency III induces IMPDH metabolism dependency

Given that XMP was one of the most highly LMP1-induced metabolites, we next investigated the effects of IMPDH1/2 inhibition by the highly selective antagonist MPA [50,51] (Fig 2A). We first cross-compared MPA effects on proliferation of LMP1-negative Burkitt cells with that of two LCLs that express LMP1 as part of the latency III program. Using the carboxyfluorescein succinimidyl ester (CFSE) dye-dilution assay, in which each cell cycle results in a 50% reduction of CFSE signal, we observed that MPA inhibited Burkitt and LCL proliferation in a dose-dependent manner. This result suggests that IMPDH activity is likely necessary for maintenance of Burkitt and LCL GTP pools. By contrast, FACS analysis of vital dye 7-Aminoactinomycin D (7-AAD) uptake revealed that MPA rapidly triggered LCL but not Burkitt cell death (Figs 2C and S2B). We therefore cross-compared isogenic Burkitt cells that differ by EBV latency program. Consistent with our LCL results, MPA triggered cell death of latency III MUTU III cells [52] to a significantly greater extent than latency I MUTU I (Figs 2D and S2C). Similarly, MPA triggered cell death to a greater extent in latency III Jijoye Burkitt cells than in its P3HR-1 subclone [53], which harbors an EBNA2 deletion and exhibits a more restrictive form of EBV latency (Figs 2E and S2D). These results suggest that latency III creates a metabolic dependency on IMPDH activity for survival.

Guanylate pool depletion by MPA can trigger apoptosis in cancer cells by inducing nucleotide imbalance and DNA damage [54,55]. We therefore tested whether MPA selectively increased DNA damage in latency III B-cells, using phosphorylation of the kinases ATM and ATR as readouts of host cell responses to double stranded versus single stranded DNA breakage, respectively. Unexpectedly, MPA lowered ATM and ATR phosphorylation levels in latency III vs latency I infected B cells (S2E Fig). Yet, MPA triggered death in MUTU III and GM12878 to a greater extent than in MUTU I, and this could be rescued by GTP supplementation, supporting the hypothesis that latency III drives an increased dependence on GTP pools for survival (Figs 2F and S3). Furthermore, MPA more highly induced executioner caspase-3 and 7 activity in MUTU III and GM12878, and this was again largely rescuable by GTP supplementation (Fig 2G). Taken together, our

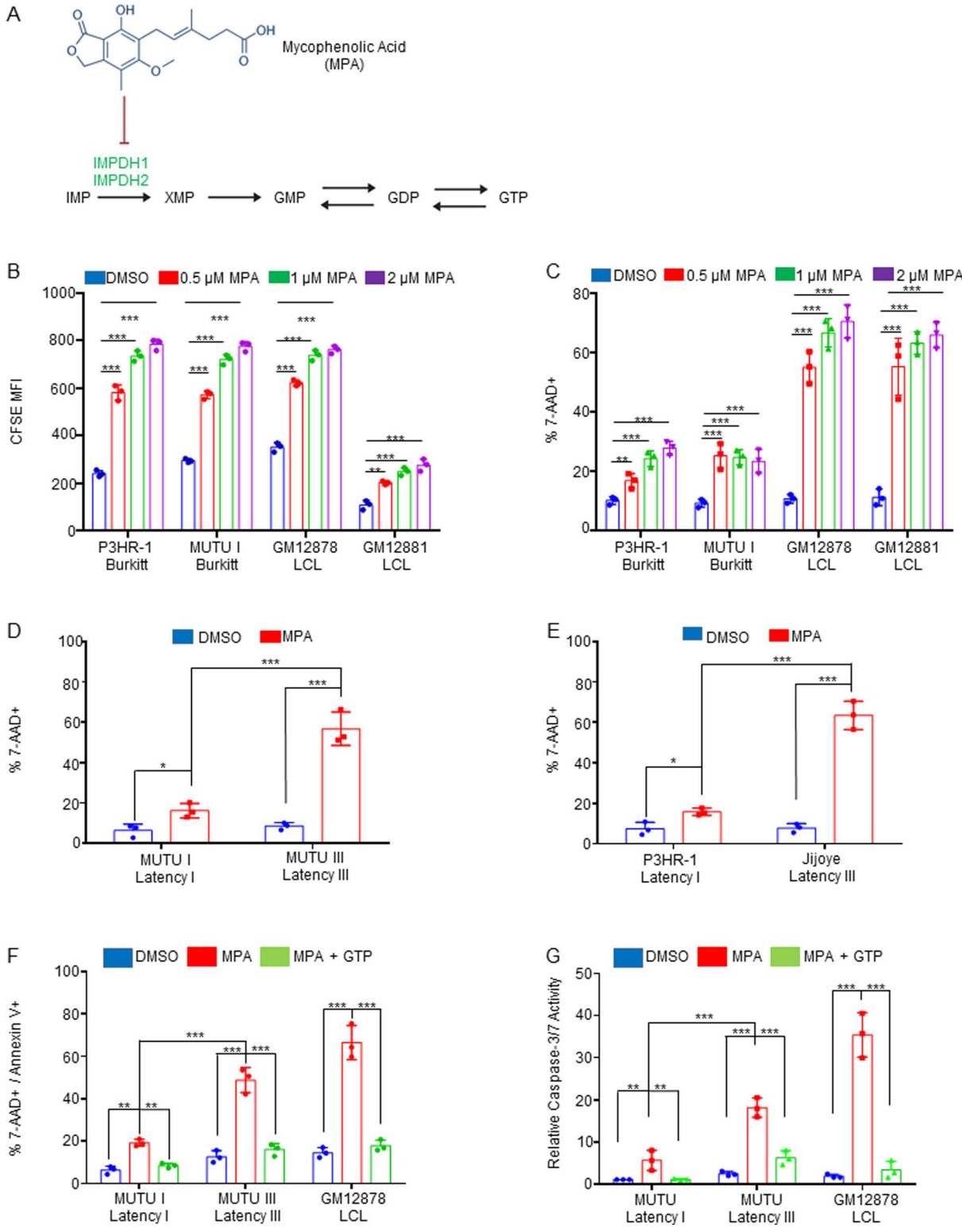

**Fig 2. EBV latency III creates dependency on IMPDH activity for prevention of apoptosis.** (A) Schematic diagram of mycophenolic acid (MPA) inhibition of the guanylate biosynthesis pathway enzymes IMPDH1 and IMPDH2. IMP, inosine monophosphate. XMP, xanthosine monophosphate, GMP,

guanosine monophosphate. GDP, guanosine diphosphate. GTP, guanosine triphosphate. (B) FACS analysis of dose-dependent MPA effects on proliferation of latency I P3HR-1 and MUTU I cells versus latency III GM12878 and GM12881 LCLs, as judged by CFSE dye-dilution analysis. CFSE-stained cells were incubated with DMSO vehicle vs the indicated MPA concentrations for 96 hours and CFSE mean fluorescence intensity (MFI) was analyzed by FACS. CFSE levels are reduced by half with each mitosis. Shown are mean ± SD CFSE levels from n = 3 independent replicates. (C) FACS analysis of dose-dependent effects of MPA treatment for 48 hours on cell death of latency I P3HR-1 and MUTU I cells versus latency III GM12878 and GM12881 LCLs, as judged by uptake of the vital dye 7-AAD. Shown are mean ± SD percentages of 7-AAD+ (non-viable) cells from n = 3 independent replicates. (D) FACS analysis of DMSO versus MPA effects on viability of isogenic MUTU I versus III cells that differ by EBV latency I versus III programs, respectively. Shown are mean ± SD percentages of 7-AAD+ cells following DMSO versus 1 µM MPA treatment for 48 hours. (E) FACS analysis of DMSO versus MPA effects on P3HR-1 versus Jijoye Burkitt cell viability following DMSO versus 1 µM MPA treatment for 48 hours. Mean ± SD 7-AAD+ cell percentages from n = 3 replicates are shown. (F) FACS analysis of mean ± SD percentages of 7-AAD + /Annexin V+ cells following treatment with DMSO, 1 µM MPA with or without 100 µM GTP rescue for 48 hours. (G) Relative caspase 3/7 activity levels of cells analyzed in panel (F), as judged by Caspase3/7 Glo assay. Mean ± SD values from n = 3 replicates are shown. *, $P<0.05$; **, $P<0.05$; ***, $P<0.005$; ns, nonsignificant using Student's t-test.

results are consistent with a model in which latency III increases both GTP demand and GTP biosynthesis, the latter of which is increased at the level of IMPDH activity.

p53 mutations are frequently present in Burkitt B-cells [56–58], while LCL have wild-type p53. Therefore, p53 activation by GTP pool depletion could potentially contribute to MPA-triggered cell death in LCLs. While MPA did not increase ATM or ATR phosphorylation in LCLs (S2E Fig), p53 could be activated in a non-canonical fashion in response to guanylate pool depletion. To investigate this, we treated two LCL, GM12878 and GM15892, with pifithrin-α, a small molecular p53 inhibitor [59], together with MPA. However, pifithrin-α p53 inhibition did not impair cell death, as judged by 7-AAD uptake and Annexin-V positivity (S4 Fig). To further exclude p53 roles, we tested MPA effects on control versus CRISPR p53 depleted GM12878 and GM15892 LCLs. In support of the pifithrin-α results, p53 depletion did not significantly alter MPA-induced LCL death (S5 and S6 Figs). Together with the observation that MPA induces MUTU III but not MUTU I death, we conclude that MPA-induced latency III B-cell death is not dependent on p53.

We next investigated the cell death pathway triggered by MPA. Since caspase-3/7 activation and Annexin V positivity can be indicative of apoptosis, we tested whether the pan-caspase inhibitor ZVAD-FMK could rescue MPA-induced death. While ZVAD-FMK inhibited MPA-induced Caspase-3/7 activity, it failed to block MPA-induced LCL death (S7 and S8 Figs). We therefore next tested whether MPA induces necroptosis, a programmed cell death pathway that can be activated in response to a range of noxious stimuli, including in the setting of apoptosis inhibition [60,61]. Yet the RIP1 inhibitor Necrostatin-1s, which blocks necroptosis initiation [62], also failed to rescue MPA-induced cell death, alone or in combination with ZVAD-FMK (S7 and S8 Figs).

## Both IMPDH isoforms are important for LCL proliferation

To define IMPDH1 versus IMPDH2 roles in latency I Burkitt versus latency III LCL proliferation, we tested effects of their CRISPR KO (Table 1). While depletion of IMPDH1 or IMPDH2 alone did not significantly alter MUTU I or Daudi Burkitt proliferation (Figs 3A, 3B, S9A and S9B), depletion of either significantly impaired proliferation of the LCLs GM12878 and GM13111, with a stronger IMPDH2 KO phenotype (Figs 3C, 3D, S9A and S9B). Whereas IMPDH2 KO significantly reduced GM12878 LCL live cell numbers, combined IMPDH1 and 2 depletion was required to reduce P3HR-1 Burkitt

**Table 1. CRISPR-Cas9 sgRNA sequences.**

| Non-targeting sgRNA | Sense | TTGACCTTTACCGTCCCGCG |
|---|---|---|
| IMPDH1 sgRNA | Sense | GTGTAGCGAGGTCCAATCTA |
| IMPDH2 sgRNA | Sense | GGCAGCCATTGGCACTCATG |
| P53 sgRNA #1 | Sense | CCATTGTTCAATATCGTCCG |
| P53 sgRNA #2 | Sense | GAGCGCTGCTCAGATAGCGA |

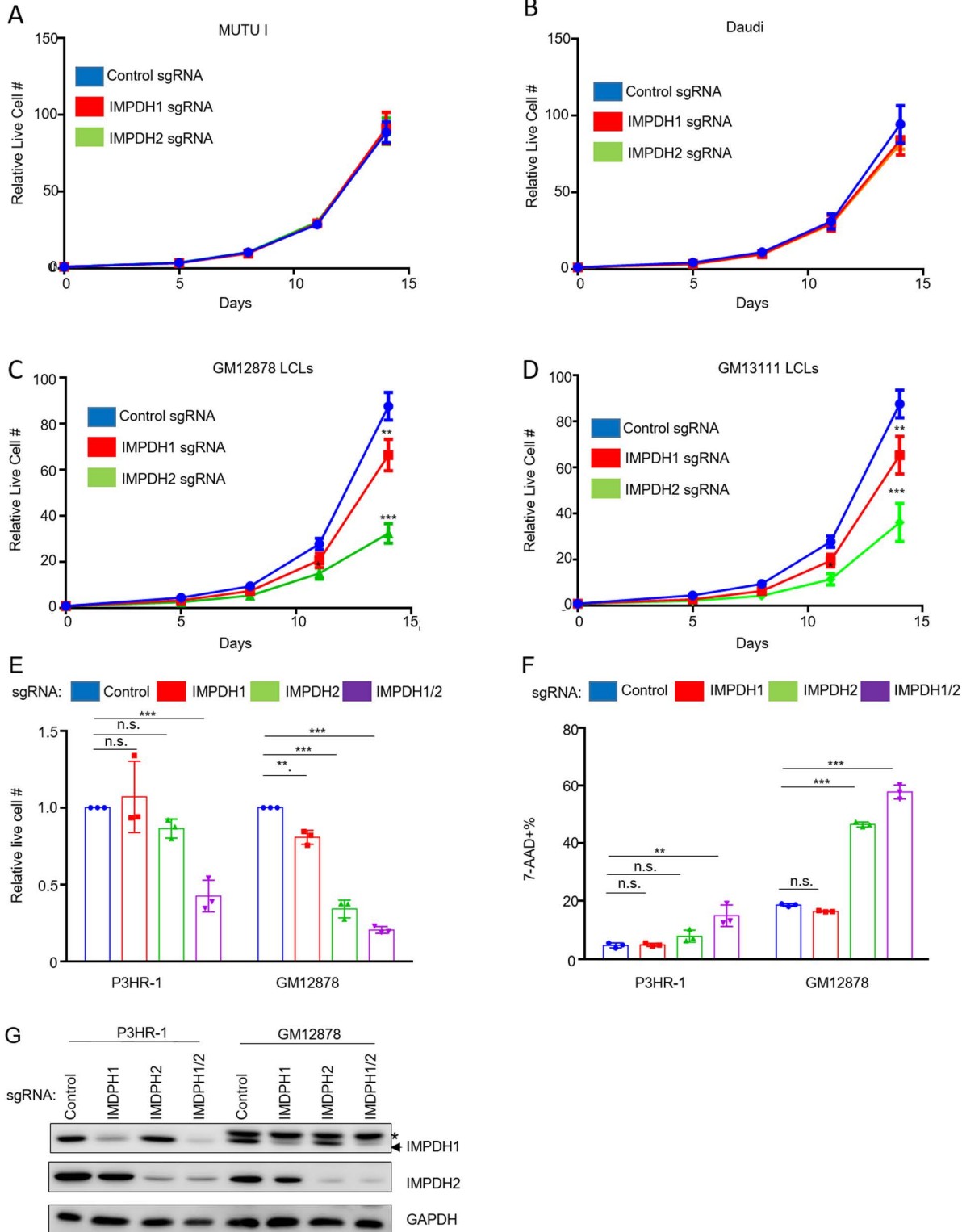

**Fig 3. LCLs but not Burkitt cells are dependent on IMPDH2 for growth and survival.** (A) Mean ± SD live cell numbers of Cas9 + MUTU I expressing control, IMPDH1 or IMPDH2 targeting single guide RNAs (sgRNA) from n = 3 replicates. Cells transduced with lentiviruses expressing the indicated sgRNAs were puromycin selected. Cell numbers immediately following puromycin selection (defined as day 0 of the graph) were set to 1. Live cell

numbers were quantitated by CellTiter-Glo assay. (B) Mean±SD live cell numbers of Cas9+Daudi cells as in (A). (C) Mean±SD live cell numbers of Cas9+GM12878 LCLs as in (A). (D) Mean±SD live cell numbers of Cas9+GM13111 LCLs as in (A). (E) Mean±SD live cell numbers of Cas9+P3HR-1 or GM12878 cells transduced with lentivirus expressing control, IMPDH1, IMPDH2 or IMPDH1 and 2 sgRNAs at 8 days post-puromycin selection. (F) Immunoblot analysis of WCL from Cas9+P3HR-1 or GM12878 expressing the indicated sgRNA. *=non-specific band present in analysis of GM12878 lysates. Immunoblots are representative of n=3 independent replicates. (G) Mean±SD MFI of Cas9+P3HR-1 or GM12878 cells transduced with lenti-virus expressing control, IMPDH1, IMPDH2 or IMPDH1 and 2 sgRNAs at 8 days post-puromycin selection, performed on cells from the same replicates shown in (E).

viability (Figs 3E–3G and S9C). By contrast, combined IMDPH1/2 depletion had only slightly greater effect on GM12878 viability than IMPDH2 depletion alone, as judged by 7-AAD uptake and Cell Titer Glo assays (Figs 3E–3G, S9D and S9E). These data are consistent with a model in which LMP1 increases IMPDH2 activity to support elevated GTP demand in latency III.

EBV induces IMPHD1 and IMPDH2 expression at the mRNA and protein levels by day two post primary human B-cell infection (S10A–S10C Fig), a timepoint at which EBNA2 is highly expressed but before LMP1 is induced [42,63–65]. Similarly, withdrawal of conditional EBNA2 expression from 2-2-3 LCLs reduced protein levels of both IMPDH1 and IMPDH2 (S10D Fig), whereas conditional Burkitt LMP1 expression did not increase IMPDH1 or 2 protein levels (S10E Fig). Furthermore, IMPDH1/2 mRNA levels were not significantly depleted by 24 hours post LMP1 CRISPR knockout (KO) in GM12878 LCLs [48], consistent with a recent report that EBNA2 and MYC but not LMP1 induce IMPDH2 [42,63–65]. Therefore, whereas EBNA2 induces IMPDH1/2 expression, LMP1 may instead increase their activity.

IMPDH activity can be boosted by the formation of large, oligomeric structures called cytoophidium observable by confocal or electron microscopy [66]. Cytoophidium are observed in murine germinal centers B-cells [67]. We therefore investigated whether potential cytoophidium roles in LMP1 upregulation of IMPDH activity in support of XMP production. Since it has been reported that MPA induces cytoophidium in a range of epithelial cells [68,69], we first treated normal oral keratinocytes (NOK) with MPA as a positive control and observed IMPDH1 and 2 cytoophidium formation (S11 Fig). We next characterized cytoophidium formation in Daudi cells, at baseline and following LMP1 expression. In contrast to NOK, we observed IMPDH1 and IMPDH2 cytoophidium in Daudi cells, which did not substantially change with 24 hours of LMP1 expression (S12 Fig). Yet, we did not observe cytoophidium in GM12878 LCLs (S13 Fig). Taken together with our observation that LMP1 increased Daudi inosine monophosphate and ribose-5-phosphate levels (Fig 1A and 1B), our data supports a model in which LMP1 instead supports purine biosynthetic pathway flux rather than driving cytoophidium formation.

### LMP1 sensitizes Burkitt cells to MPA-induced cell death in a GTP-dependent manner

We next tested the hypothesis that LMP1 itself creates IMPDH dependency. We conditionally expressed LMP1 in Daudi cells, in the absence or presence of MPA. LMP1 signaling was not blocked by MPA, as judged by processing of the non-canonical NF-κB pathway precursor p100 into the active p52 product (Fig 4A). MPA induced significantly higher cell death levels in LMP1+ than in LMP1- Daudi cells, as judged by 7-AAD uptake (Fig 4B and 4C), indicating that LMP1 signaling creates a metabolic dependence on IMPDH activity for B-cell survival.

To investigate whether signaling from the TES1 and/or TES2 domains were responsible for LMP1-driven IMPDH dependency, we conditionally expressed wildtype LMP1 or point mutants abrogated for signaling by TES1 (TES1 mutant, TES1m), TES2 (TES2m), or both (TES1m+TES2m) [70–72]. We validated that the TES1 point mutation abrogated non-canonical NF-κB pathway p100 to p52 processing as expected (Fig 4D). Intriguingly, MPA more highly induced cell death in cells with wildtype than with either TES1m or TES2m LMP1 expression (Figs 4E, 4F, and S14). By contrast, expression of the TES1m+TES2m double LMP1 point mutant did not sensitize Daudi cells to MPA-driven killing (Figs 4E, 4F, and S14). These results indicate that signaling by either TES1 or TES2, rather LMP1 expression itself, contribute to

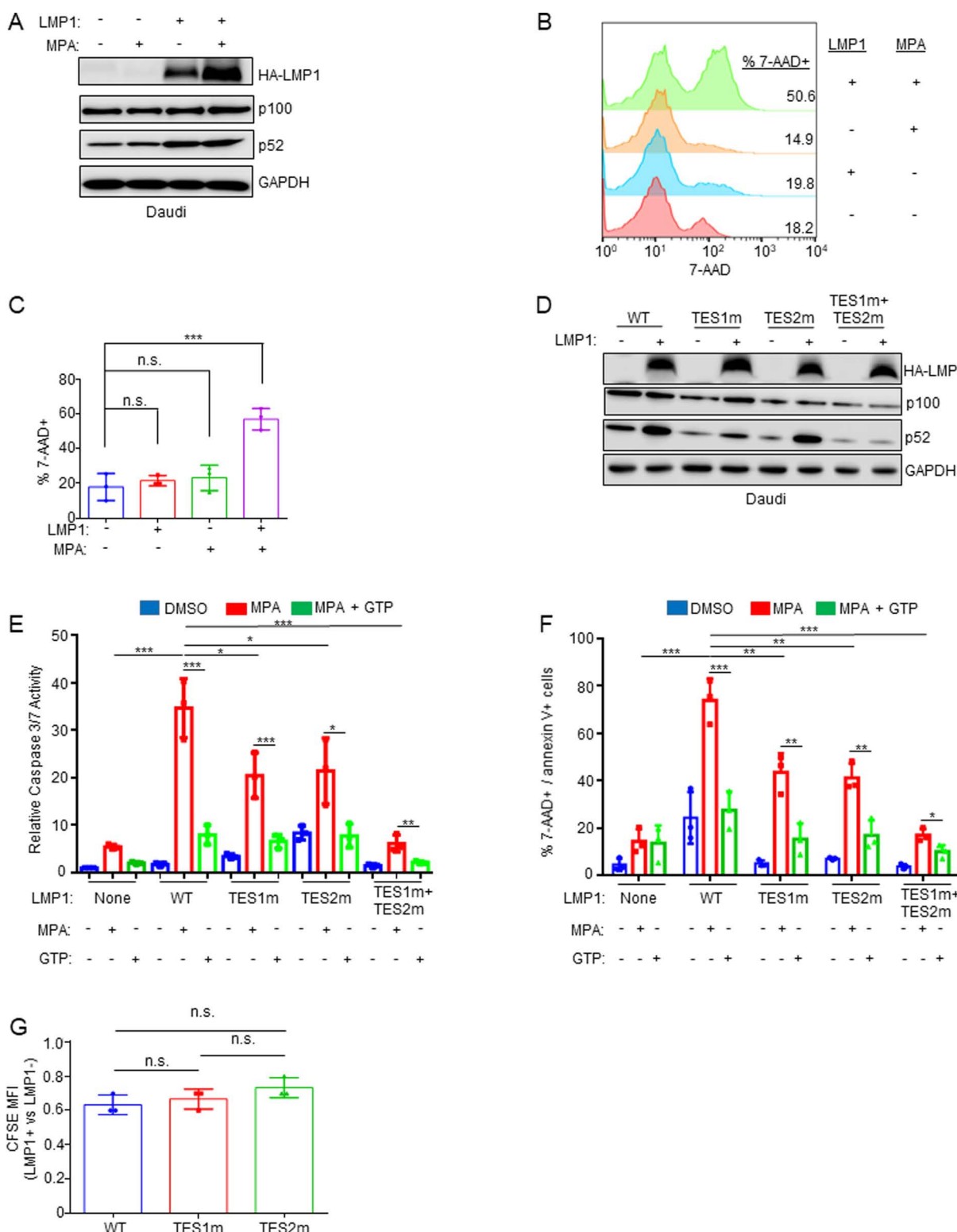

**Fig 4. LMP1 expression sensitizes Burkitt cells to MPA-driven death in a partially GTP dependent manner.** (A) MPA does not impair conditional LMP1 expression or signaling. WCL of Daudi cells induced for LMP1 and/or treated with MPA 1 μM for 24 hours, as indicated. (B) Analysis of LMP1 effects on MPA-driven Burkitt cell death. FACS analysis of 7-AAD uptake by Daudi cells mock induced or induced for LMP1 and then treated with DMSO

vs MPA for 96 hours. Shown at right are the % 7-AAD+ cells under each condition. (C) Mean±SD percentages of 7-AAD cells from n = 3 replicates as in (B). (D) Validation of Daudi conditional wildtype vs mutant LMP1 expression. Immunoblot analysis of WCL of Daudi cells induced for wildtype (WT), TES1m, TES2m or TES1+TES2m LMP1 for 24 hours. (E) Analysis of TES1 vs TES2 effects on sensitization to MPA-induced Burkitt apoptosis. Relative caspase-3/7 activities in Daudi cells mock induced or induced for WT or the indicated LMP1 mutant for 24 hours and then treated with 1 μM MPA±100 μM GTP rescue for 96 hours. Caspase 3/7 levels were measured by Caspase-3/7 Glo assay, and values in mock induced and DMSO treated cells were set to 1. (F) Analysis of TES1 vs TES2 effects on sensitization to MPA-induced Burkitt apoptosis. Mean±SD percentages of double 7-AAD+/Annexin V+ from n = 3 replicates of Daudi cells treated as in (E). Immunoblots are representative of n = 3 replicates. *P < 0.05, **P < 0.01, ***P < 0.005.

dependence on IMPDH for survival. Similar results were obtained in EBV-negative BL-41 Burkitt cells, where MPA treatment again induced significantly higher death in cells with wildtype than TES1 or TES2 mutant LMP1 expression. These phenotypes were partially rescuable by GTP supplementation (S15 and S16 Figs), suggesting that TES1 and TES2 signaling each increase GTP biosynthesis to meet increased demand.

## TES1 and TES2 each remodel B-cell metabolism

To gain further insights into TES1 and TES2 metabolism remodeling roles including at the nucleotide biosynthesis level, we performed LC/MS profiling of Burkitt cells at 24 hours post LMP1 TES1m versus TES2m expression. Samples were run in the same biological replicates as ones from cells mock induced or induced for wildtype LMP1 expression (Fig 1) to facilitate cross-comparison and to limit batch effects. Interestingly, expression of TES1 mutant LMP1 (which signals from TES2) only mildly upregulated XMP and downregulated GTP abundance, relative to levels observed in mock induced Daudi cells (Fig 5A). TES2 mutant LMP1 (which signals from TES1) instead upregulated XMP abundance by ~2-fold and significantly increased GMP levels (Fig 5B). TES2 mutant LMP1 likewise upregulated Akata XMP levels by ~4-fold, whereas levels were again only modestly increased by TES1 mutant LMP1 (S17A and S17B Fig). These data suggest that TES1 signaling may more strongly induce IMPDH activity and/or that TES2 signaling more strongly increases XMP demand.

We next cross-compared metabolite abundances in Daudi cells expressing TES1 mutant versus wildtype LMP1. Wildtype LMP1 induced significantly higher levels of the GMP/AMP precursor inosine monophosphate (IMP), GDP and GTP than TES1 mutant LMP1. Deoxyinosine was also significantly higher in both Daudi and Akata expressing wildtype than TES1 mutant LMP1 (Figs 5C and S17C), further implicating TES1 signaling in nucleotide metabolism remodeling. By contrast, GDP or GMP levels were not significantly different in cells expressing TES2 mutant versus wildtype LMP1 (Figs 5D and S17D). Taken together, these results indicate that TES1 signaling may more strongly induce purine synthesis to support GTP abundance.

## LMP1 primarily utilizes PKC signaling to regulate purine metabolism

To characterize pathways downstream of LMP1 important for upregulation of XMP abundance and of guanosine metabolism more generally, we performed LC/MS metabolomic analyses in LMP1-expressing Daudi cells treated with DMSO vehicle versus with a panel of well-characterized chemical inhibitors. LMP1 was doxycycline induced for 24 hours in the presence of DMSO, the protein kinase C inhibitor (PKCi) staurosporine (100 nM), JNK inhibitor (JNKi) SP600125 (10 μM), ERK inhibitor (ERKi) SCH772984 (10 μM), p38 inhibitor (p38i) adezmapimod (10 μM) or IKKβ inhibitor (IKKβi) IKK-2 VII (1 μM). We validated that these inhibitors did not significantly change cell viability at the early 24 hour timepoint in LMP1 expressing cells (Fig 6A). We performed LC/MS metabolomic analysis using n = 4 replicates, which highlighted that each inhibitor significantly altered the metabolome in LMP1-expressing Daudi (Fig 6B and S3 Table). Intriguingly, each significantly decreased levels of the XMP precursor IMP as well as of XMP itself, suggesting that LMP1-activated PKC, MAP kinase and NF-kB pathways each support XMP metabolism (Fig 6C and 6D). However, whereas PKC inhibition also strongly diminished levels of GMP, GDP and GTP relative to those observed in DMSO-treated cells, the other inhibitors failed to do so (Fig 6C and 6D). Notably, levels of pyrimidine nucleotides were not significantly changed by these inhibitors. Collectively, these studies highlight roles of multiple LMP1 activated pathways in support of XMP production, as well as PKC roles in support of GTP pools.

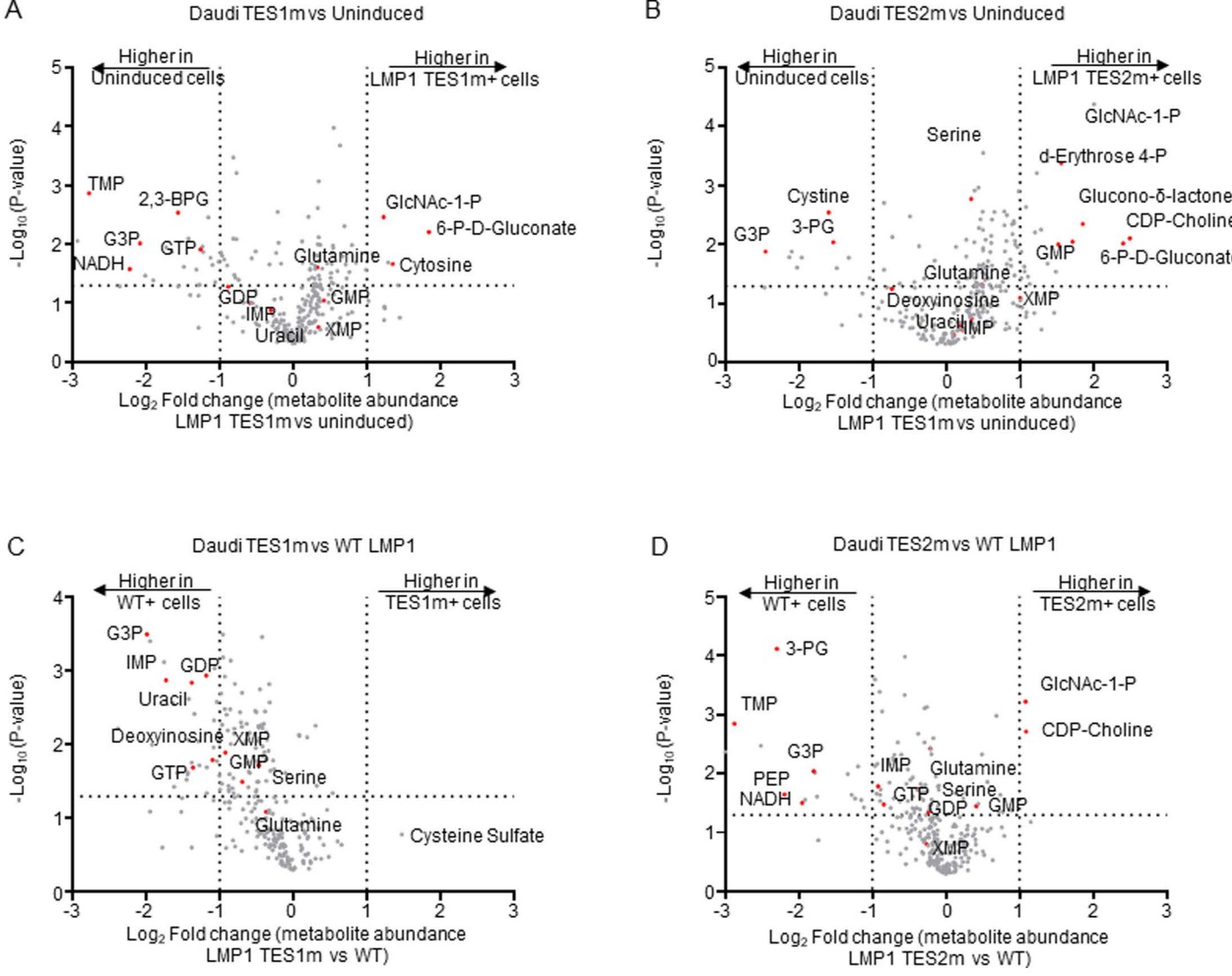

**Fig 5. Effects of LMP1 TES1 versus TES2 signaling on Daudi Burkitt metabolome remodeling.** (A) Volcano plot of LC-MS metabolomic analysis of n = 3 replicates of Daudi cells mock induced or doxycycline induced for LMP1 TES1m expression for 24 hours. Metabolites with higher abundance in LMP1 TES1+ cells have positive fold change values, whereas those higher in mock induced cells have negative fold change values. Selected metabolites are highlighted by red circles and annotated. (B) Volcano plot of LC-MS metabolomic analysis of n = 3 replicates of Daudi cells mock induced or doxycycline induced for LMP1 TES2m expression for 24 hours, with selected metabolites highlighted as in (A). (C) Volcano plot of LC-MS metabolomic analysis of n = 3 replicates of Daudi cells doxycycline induced for TES1m vs WT LMP1 expression for 24 hours, with selected metabolites highlighted. Replicates for this cross-comparison were induced side by side, prepared for and analyzed by LC-MS together on the same day to minimize batch effects. (D) Volcano plot of LC-MS metabolomic analysis of n = 3 replicates of Daudi cells doxycycline induced for TES2m vs WT LMP1 expression for 24 hours, with selected metabolites highlighted. Replicates for this cross-comparison were induced side by side, prepared for and analyzed by LC-MS together on the same day to minimize batch effects.

## IMPDH metabolism crosstalk with EBV epigenome heterochromatin

We noted that MPA treatment induced Burkitt cell homotypic adhesion, in which large clusters or MPA-treated MUTU I or Daudi cells were evident (Fig 7A). Since LMP1 induces B-cell homotypic aggregation likely by upregulating plasma membrane L-selectin and other adhesion molecules [71,73], we investigated whether MPA de-repressed Burkitt LMP1

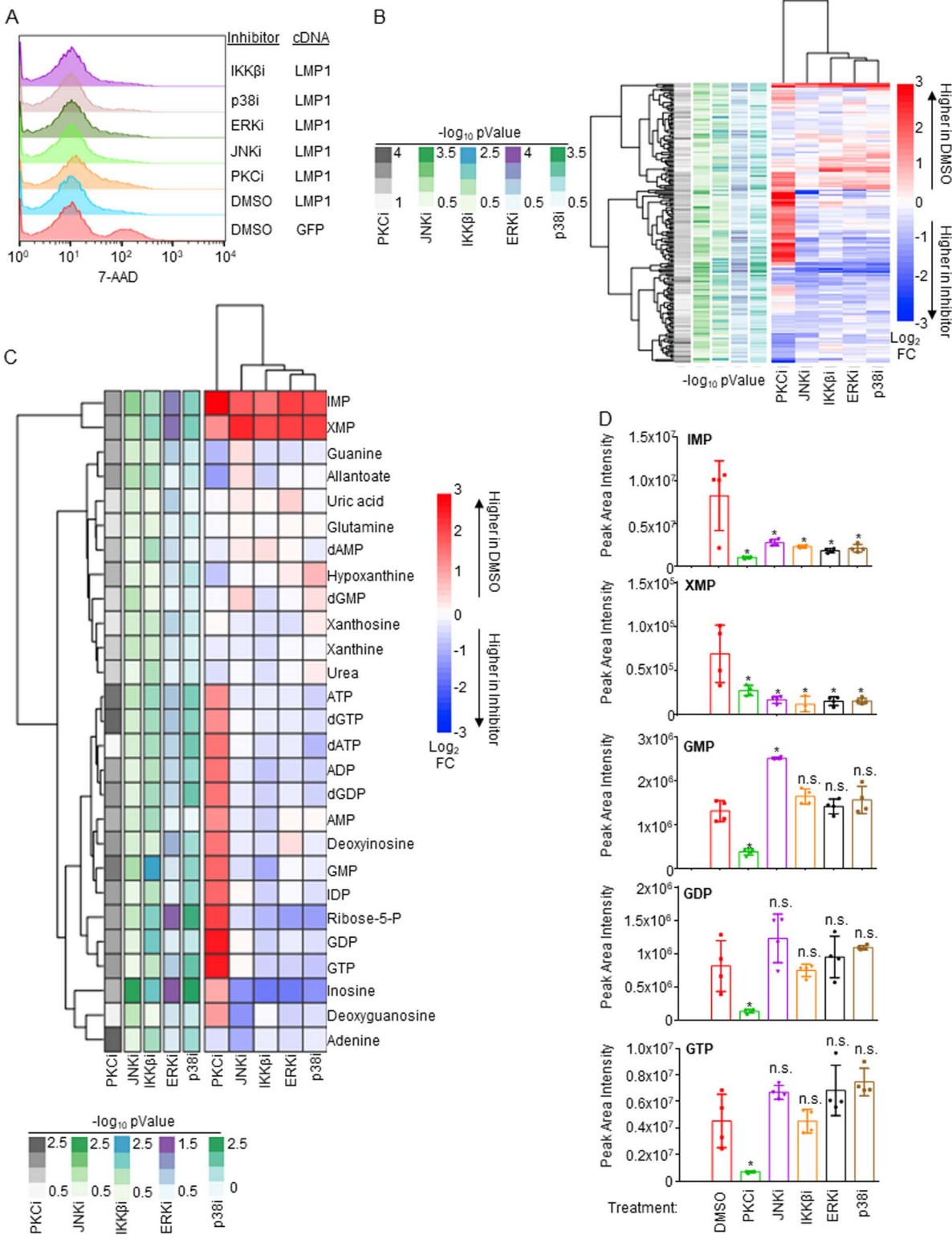

**Fig 6. Analyses of LMP1 pathway inhibition effects on Burkitt B cell metabolome remodeling.** (A) FACS analysis of 7-AAD uptake in Daudi cells conditionally induced for control GFP versus LMP1 expression for 24 hours, in the presence of DMSO vehicle or of the indicated small molecule

inhibitors: PKC inhibitor (PKCi) staurosporine (100nM), JNK inhibitor (JNKi) SP600125 (10 µM), ERK inhibitor (ERKi) SCH772984 (10 µM), p38 inhibitor (p38i) adezmapimod (10 µM) or IKKβ inhibitor (IKKβi) IKK-2 VII (1 µM). None of the inhibitors increased cell death at the early 24 hour timepoint in LMP1 expressing cells. (B) LC/MS metabolomic analyses was performed using n = 4 independent replicates on Daudi cells treated as in (A). Shown is a heatmap depicting significantly changed metabolite abundances in Daudi cells treated with the indicated inhibitor versus in DMSO treated controls, using a p < 0.05 cutoff. $-\log_{10}$ p-Value and $\log_2$ (metabolite fold changes) scales are shown to the left and right of the heatmap, respectively. (C) Heatmap of significantly changed purine metabolism pathway metabolites (using a p < 0.05 cutoff). $-\log_{10}$ p-Value and $\log_2$ (fold change metabolite) scales are shown below and to the right of the heatmap, respectively. (D) Bar graphs showing LC-MS mean ± SD of peak area intensities from n = 4 replicates of the indicated guanylate biosynthesis pathway metabolites. Statistical significance of changes between inhibitor treated vs DMSO treated samples is indicated. * = p < 0.05, n.s. = not significant.

expression. Immunoblot analysis demonstrated LMP1 expression in MPA-treated MUTU I, Akata, and Rael Burkitt cells, which could be suppressed by GTP supplementation (Fig 7B). Since Daudi harbor an EBV genomic deletion that removes *EBNA2*, MPA effects on LMP1 were not indicative of a switch to the latency III program, and EBNA2 was not de-repressed in MUTU I, which have intact EBV genomes (S18A Fig). LMP2A was also not de-repressed by MPA in either Daudi or MUTU I (S18A Fig).

Since LMP1 can be expressed as a latency gene or during the EBV lytic cycle [74,75], we tested whether MPA de-repressed other EBV lytic cycle antigens. Immunoblot analysis demonstrated that MPA de-repressed immediate early BZLF1 and early BMRF1 in MUTU I, Akata and Rael (Fig 7B). By contrast, lytic cycle antigens were not appreciably de-repressed by MPA in Daudi cells. MPA effects on lytic cycle and LMP1 expression were rescuable by GTP supplementation (Fig 7B). These data are consistent with a model in which MPA de-represses LMP1 through Burkitt cell lytic reactivation.

To gain insights into MPA effects on the EBV epigenome, we performed chromatin immunoprecipitation (ChIP) and qPCR analysis. MPA reduced repressive histone 3 lysine 9 trimethyl (H3K9me3) marks and increased activating H3K9 acetyl (H3K9ac) marks at both the LMP1 and BZLF1 promoters in MUTU I cells (Table 1 and Fig 7C and 7D). MPA also increased repressive histone 3 lysine 27 trimethyl (H3K27me3), but did not substantially alter H3K27 acetyl (H3K27ac) mark abundance at the LMP1 promoter in Daudi cells (Table 1 and S18B Fig). MPA epigenetic effects were rescuable by GTP supplementation (Fig 7C and 7D). MPA induced similar epigenetic remodeling at the EBV genomic Q promoter, active in latency I, and at the latency III C promoter, repressed in latency I (Table 1 and Fig 7E and 7F). These results indicate that IMPDH activity broadly supports the latency I EBV epigenome rather than exerting specific effects at the LMP1 promoter.

EBV lytic activation promotes a pseudo-G1/S state [76]. We therefore investigated MPA effects on Burkitt cell cycle. Using propidium iodide (PI) versus 5-ethynyl-2′-deoxyuridine (EdU) staining, we found that MPA induced G1 arrest (S18C Fig). We therefore next tested whether MPA triggered a full EBV lytic cycle with secretion of encapsidated EBV virions. Interestingly however, viral load analysis identified that MPA treatment did not trigger EBV secretion (Table 2 and S18D Fig), suggesting that it instead induce an abortive lytic cycle. Taken together, our observations suggest that latency III B-cells depend upon LMP1-driven *de-novo* guanylate production for survival, whereas IMPDH1/2 activity instead maintains latency I Burkitt GTP levels for proliferation and latency maintenance (Fig 8).

## Discussion

Metabolism remodeling is a cancer hallmark, yet much has remained unknown about how the key EBV oncogene LMP1 remodels host B-cell oncometabolism pathways. Here, LC/MS profiling highlighted that LMP1 significantly upregulates nucleotide metabolism pathways, in particular purine metabolism. While LMP1 did not alter IMPDH1 or IMPDH2 protein levels, it significantly upregulated levels of the IMPDH product XMP. IMPDH blockade triggered Burkitt growth arrest, but instead triggered cell death in latency III Burkitt cells or LCLs, highlighting cross-talk between EBV oncoproteins and XMP metabolism. LMP1 expression itself was sufficient to create dependence on IMPDH activity for survival. IMPDH2 played a larger role in LCL proliferation, though IMPDH1 also contributed. TES1 and TES2 signaling additively contributed to dependence on IMPDH activity for B-cell survival.

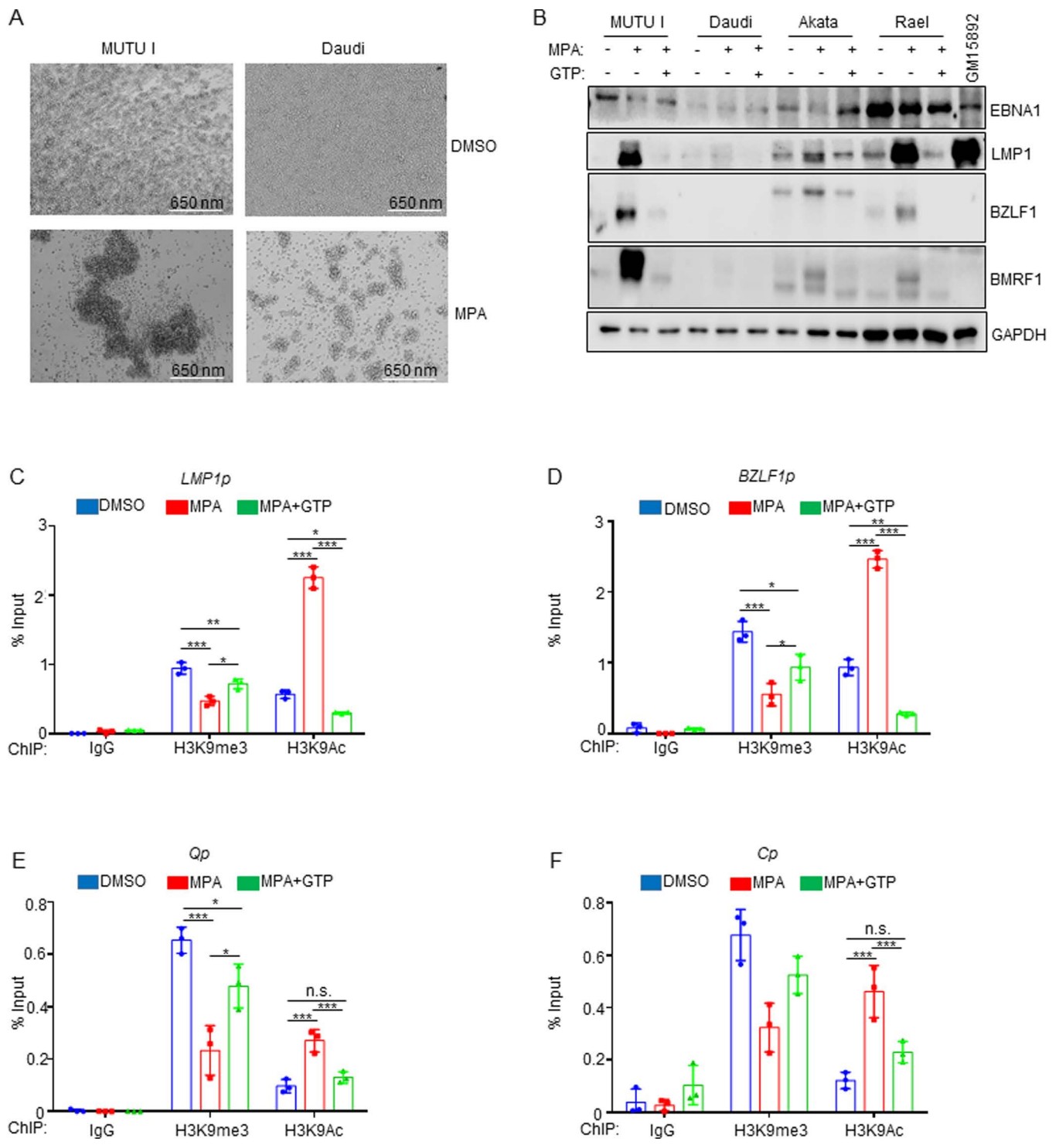

**Fig 7. MPA derepresses lytic gene expression including LMP1 in latency I Burkitt cells in a GTP dependent manner.** (A) Analysis of MPA effects on latency I Burkitt cell homotypic adhesion (cell clumping), a phenotype that typically correlates with LMP1 expression. Representative brightfield microscopy images of MUTU I and Daudi cells treated with DMSO or the indicated MPA concentration for 72 hours indicating MPA-induced homotypic adhesion. (B) Analysis of MPA effects on Burkitt LMP1, EBNA1, immediate early BZLF1 and early lytic BMRF1 expression. Immunoblot analysis of WCL from the indicated Burkitt cells mock treated or treated with 1 μM MPA± 100 μM GTP for 24 hours. Shown in the rightmost lane are WCL from latency III GM15892 LCLs for cross comparison. MPA effects on LMP1 and on lytic gene expression were largely reversed by GTP supplementation. Blots are

representative of n = 3 replicates. (C) Mean ± SD percentage of input values from MUTU I Burkitt cell chromatin immunoprecipitation (ChIP) with qPCR analysis, using the indicated control IgG, anti-H3K9me3 or anti-H3K9Ac antibodies, as indicated and with primers specific for the LMP1 promoter region (LMP1p). (D) Mean ± SD percentage of input ChIP-qPCR values as in (C) using primers specific for the immediate early BZLF1 promoter (BZLF1p) region. (E) Mean ± SD percentage of input ChIP-qPCR values as in (C) using primers specific for the latency I Q promoter (Qp) region. (F) Mean ± SD percentage of input ChIP-qPCR values as in (C) using primers specific for the latency III C promoter (Cp) region. *, P < 0.05; **P < 0.01, ***P < 0.005, ns = non-significant.

**Table 2. ChIP-qPCR primer sequences.**

| Primer | Sequence | Source |
| --- | --- | --- |
| LMP1p FP | TCCAGAATTGACGGAAGAGGTT | PMID: 28794029 |
| LMP1p RP | GCCACCGTCTGTCATCGAA | PMID: 28794029 |
| BZLF1p FP | GCAAGGTGCAATGTTTAGTGAGTT | PMID: 28794029 |
| BZLF1p RP | GCTGGTGCCTTGGCTTTAAAG | PMID: 28794029 |
| Qp FP | AAATTGGGTGACCACTGAGGGAGT | PMID: 28794029 |
| Qp RP | ATAGCATGTATTACCCGCCATCCG | PMID: 28794029 |
| Cp FP | GGCGGGAGAAGGAATAACG | PMID: 28794029 |
| Cp RP | CTTGAGCTCTCTTATTGGCTATAATCC | PMID: 28794029 |

IMPDH regulates flux at the branch point between adenine and guanine nucleotide production and is the rate-limiting step in guanine nucleotide biosynthesis [77]. A future objective will be to identify why LMP1 and latency III create metabolic dependency on IMPDH-driven GTP biosynthesis for B-cell survival, even in the context of its conditional Burkitt expression. Whereas lymphocytes typically rely on *de novo* biosynthesis to meet GTP demand, MPA does not induce B-cell apoptosis in most contexts, including of primary human peripheral blood or tonsil naïve, memory B-cells or plasmablasts [78]. Interestingly, despite the similarity between LMP1 and CD40 signaling pathways [10,11], MPA also does not trigger death of primary human B-cells stimulated for 24 hours by CD40-ligand together with anti-IgM cross-linking and interleukin-21 [78]. However, ex vivo stimulation using CD40-ligand and anti-IgM stimulation is distinct from germinal center physiology. We hypothesize that LMP1 signaling pathway(s) not shared with CD40 markedly increase GTP demand, necessitating increased IMPDH2 and to a lesser extent IMPDH1 activity for B-cell survival.

Consistent with a recent report that EBNA2 induces IMPDH2 expression [42,63–65], we did not find evidence for induction of IMPDH1 or IMPDH2 mRNA or protein expression by LMP1. Since LMP1 instead increased IMPDH product XMP levels, our results suggest that LMP1 instead increases IMPDH activity, perhaps at the level of post-translational modification. While accumulation of the IMPDH precursor IMP drives formation of cytoophidium [79–82], which reduce allosteric IMPDH1 and IMPDH2 inhibition by GTP and ATP [79,80,83,84], we did not obtain evidence that LMP1 increases cytoophidium formation. Alternatively, IMPDH1 can be phosphorylated on at least three sites, including by protein kinase Cα (PKC) [85]. Notably, LMP1 TES1/CTAR1 signaling activates PKC family members [86,87], and we found that PKC inhibition strongly reduced XMP, GMP, GDP and GTP levels. We also identified major roles of MAP kinase and canonical NF-κB in support of XMP abundance in LMP1-expressing cells, but their inhibition did not reduce GTP levels, perhaps due to reduced consumption. It will be of interest to determine whether LMP1 alters IMPDH phosphorylation, an alternative post-translational modification, or a protein-protein interaction to increase IMPDH activity. Alternatively, we observed increased ribose-5-phosphate and IMP abundance with LMP1 expression, and it remains possible that LMP1 regulates nucleotide precursor flux through IMPDH to promote nucleotide biosynthesis.

Vertebrates encode two IMPDH enzymes, IMPDH1 and IMPDH2. Point mutations in each are linked to distinct human diseases [77]. Our CRISPR analysis found that both IMPDH1 and IMPDH2 were important for latency I Burkitt B-cell proliferation, but IMPDH2 was more important for LCL survival than IMPDH1. Whereas IMPDH1 is widely expressed

PLOS Pathogens

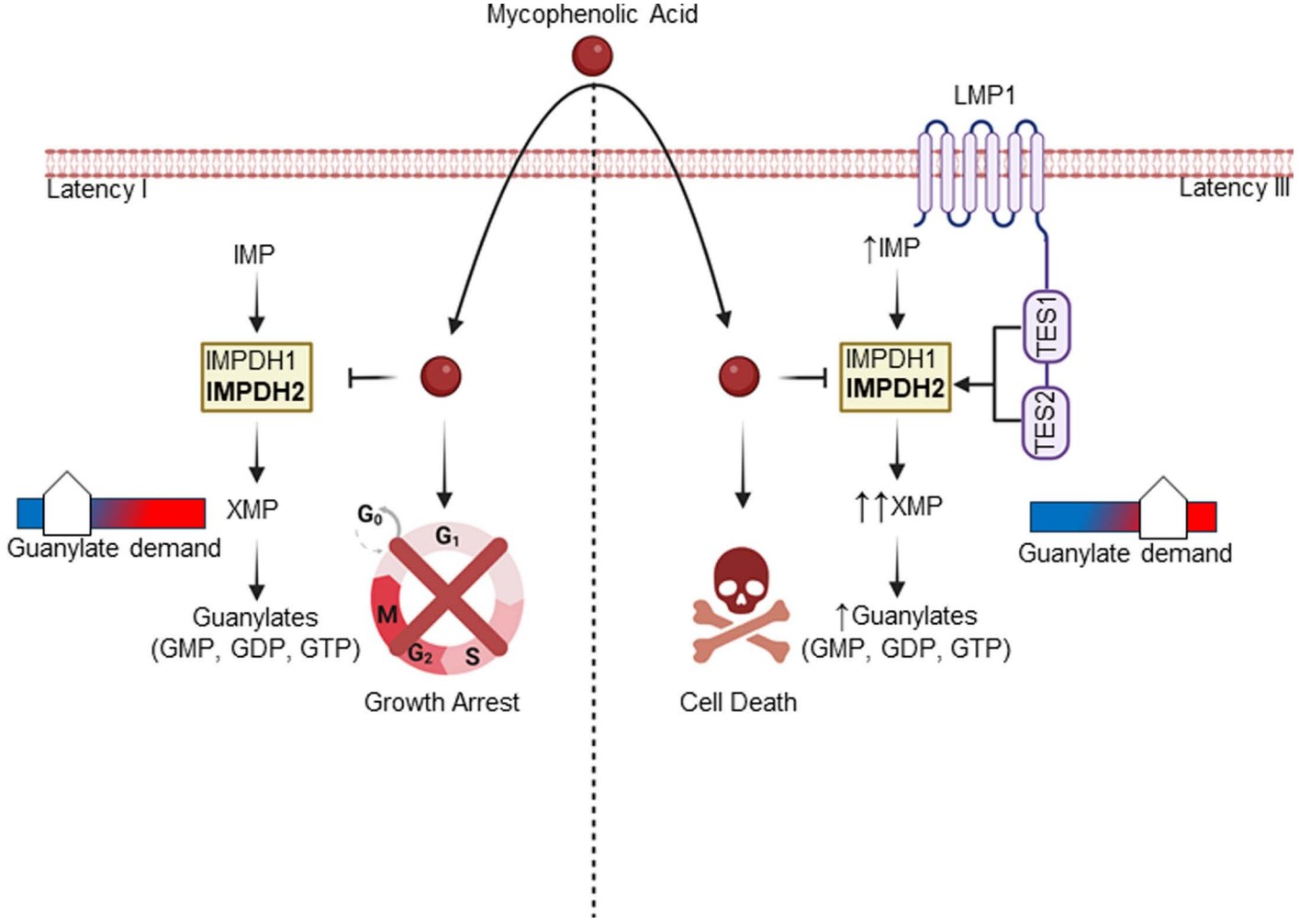

**Fig 8. Schematic model of IMPDH1/2 inhibition effects on LMP1-negative latency I versus LMP1-positive latency III B-cell growth and survival.** In Latency I, IMPDH1 and 2 each contribute to production of XMP and downstream guanylates to support demand. IMPDH1/2 inhibition by MPA triggers Burkitt growth arrest and de-represses EBV lytic antigens. In Latency III, LMP1 activated IMPDH2 predominantly supports XMP production and guanylate synthesis, sensitizing LMP1-expressing cells to MPA-driven killing. Created in BioRender. Burton, E. (2025) https://BioRender.com/76p9mp7.

across many tissues, IMPDH2 is more abundantly expressed in activated lymphocytes [78,88] and is nearly 5-fold more susceptible to MPA inhibition than IMPDH1 [89]. We therefore speculate that LMP1 expression necessitates upregulation of IMPDH2 activity to meet the elevated GTP demand, in the absence of which cell death is triggered. Development of IMPDH-isoform-specific inhibitors could provide therapeutic strategies against latency III/LMP1 expressing cancers with less side effects.

What then triggers cell death upon IMPDH inhibition of LMP1 + B-cells? p53 drives apoptosis in response to GTP depletion in renal cells [90], although EBNA3C downmodulates p53 function in latency III [91,92]. Nonetheless, LCLs have wildtype p53, and increases in p53 abundance such as by MDM2 perturbation can trigger apoptosis [93]. We therefore investigated potential p53 executioner roles, but found that p53 inhibition by CRISPR or chemical approaches did not rescue viability of MPA-treated LCLs, and caspase inhibition also failed to rescue MPA-triggered cell death. It remains plausible that MPA instead induces LCL necrosis, which would be consistent with the observation that MPA induces necrosis of

activated lymphocytes. This latter pathway involves activation of the Rho-GTPase Cdc42 and actin polymerization, which will be investigated in future LCL studies [94].

A prior study used a large panel of LCLs to investigate the relationship between B-cell gene expression and MPA sensitivity [95]. The four validated hits included a subunit of ribonucleotide reductase, which converts ribonucleotides into deoxyribonucleotides. The EBV-upregulated [63] iron reductase CYBRD1 was also implicated. It will therefore be of interest to investigate whether LMP1 alters the activity of these host factors. Additionally, it remains plausible that rapid changes in GTP abundance may trigger a GTP-dependent cell death pathway [96] or alter GTP-dependent trafficking of key dependency factors. For instance, we previously identified that LMP1 itself requires trafficking by the small GTPase Rab13 for effects on target gene regulation [41].

LMP1 is expressed in a subset of epithelial tumors, in particular nasopharyngeal carcinoma (NPC) [97–99]. While specific NPC IMPDH roles remain unstudied, it is notable that IMPDH2 is expressed in NPC cell lines and in tumor tissues [100]. Interestingly, elevated IMPDH2 expression can serve as an independent prognostic biomarker for poor NPC prognosis in patients with localized or advanced metastatic disease [70]. While this study did not specifically examine LMP1 expression in NPC tumor tissues, it would be of interest to know whether cases with elevated IMPDH2 levels also exhibited abundant LMP1 expression. Relatedly, an important future objective will be to identify how LMP1 expression remodels purine nucleotide biosynthesis pathways, including GTP, in epithelial cells including in the nasopharyngeal and gastric carcinoma contexts. It will also be of interest to determine whether LMP1 expression creates epithelial cell metabolic dependency on IMPDH1 and/or 2, or whether this phenotype is specific to B-cells.

IMPDH2 is over-expressed in a range of cancers including glioblastoma, where its elevated levels drive increased synthesis of rRNA and tRNA, stabilization of the GTP-binding nucleolar protein nucleostemin and nucleolar hypertrophy [101]. Increased nucleolar size was observed by two days post-EBV infection of primary human B-cells, which reached peak size at day 4 post-infection, a time-point where there are highly elevated levels of EBNA2 and MYC. MPA IMPDH inhibition reduced EBV-driven nucleolar hypertrophy and depleted nucleostemin at day 4 post-EBV infection [65]. Since LMP1 begins to be induced at the protein level by this timepoint [65], it will be of interest to determine whether TES1 or TES2 signaling increases IMPDH2 activity and XMP abundance at this early timepoint, as well as to define the earliest timepoint post-EBV infection at which LMP1 expression reaches the threshold at which infected cells become sensitized to MPA-driven cell death.

MPA inhibits the outgrowth of newly infected primary human B-cells, reduces size of EBV-transformed cord blood B-cell xenografts *in vivo* and increases survival of mice with EBV-transformed B-cell xenograft tumors [65,102]. Notably, MPA treatment greatly reduced LMP1 expression in surviving xenograft tumor cells [65], which we speculate occurred as a result of strong pro-necrosis selective pressure generated by IMPDH inhibition in cells that retained LMP1 expression. It will be of interest to define how LMP1 is silenced in this setting, presumably by EBV epigenomic changes potentially including DNA methylation or polycomb repressive complex II activity [103]. Likewise, the MPA prodrug mycophenolate mofetil (MMF) is commonly used as part of immunosuppressive regimens post-transplant or with autoimmunity. It will be of interest to define whether EBV-driven lymphoproliferative diseases that break through IMPDH antagonists in patient populations exhibit diminished LMP1 expression, and if so, whether synthetic lethal approaches can be devised to selectively target such adapted EBV-infected B-cells.

In summary, LMP1 expression remodels host B-cell nucleotide *de novo* guanine nucleotide metabolism. LMP1 increased abundance of the IMPDH product XMP and created metabolic dependency on IMDPH activity for B-cell survival. IMPDH2 played a more important role in LCL survival, though both IMPDH isoforms contributed to cell death blockade. Both TES1 and TES2 signaling supported guanine metabolism. Whereas IMPDH blockade triggered LCL death, it instead caused Burkitt growth arrest and LMP1 de-repression in the context of lytic reactivation. Our results further highlight IMPDH metabolic dependency as a rational therapeutic target for the treatment of EBV-driven immunoblastic lymphomas driven by LMP1.

## Materials and methods

### Ethics statement

Platelet-depleted venous blood obtained from the Brigham & Women's hospital blood bank were used for primary human B cell isolation, following our Institutional Review Board-approved protocol for discarded and de-identified samples. The Mass General Brigham Hospital Institutional Review Board (IRB) approved this study. Study approval number is: 2022P001270. Formal consent was obtained by the Brigham & Women's hospital blood bank during before donation.

### Cell lines and culture

293T, Daudi and Jijoye were purchased from American Type Culture Collection. P3HR-1, Daudi, Kem I, and EBV- Akata [104] were obtained from Elliott Kieff. GM11830, GM12878, GM13111, and GM12881 LCL were obtained from Coriell. MUTU I and MUTU III were obtained from Jeff Sample and Alan Rickinson. 2-2-3 LCLs with a conditional EBNA2-HT allele were obtained from Bo Zhao and Elliot Kieff. Normal Oral Keratinocytes (NOK) were obtained from Bo Zhao via Karl Munger's group. All B-cell lines were cultured in RPMI-1640 (Invitrogen) supplemented with 10% fetal bovine serum (FBS). 293T cells were cultured in DMEM with 10% FBS. NOK were maintained in Keratinocyte Serum-Free Media (Thermo Fisher, 17005042). All cell lines were incubated with 1% penicillin-streptomycin (Gibco) in a humidified incubator at 37°C and 5% CO2. All cells were routinely confirmed to be mycoplasma-negative by Lonza MycoAlert assay (Lonza). For 2-2-3 cell assays, cells cultured in the presence of 4-hydroxy tamoxifen (4HT) exhibit latency III. The conditional EBNA2-HT localizes to the nucleus in the presence of 4HT, but upon 4HT withdrawal, it is sequestered in the cytosol. 2-2-3 LCLs were maintained in the presence of 1 μM 4HT. For 4HT removal, cells were washed five times with 4HT-free media. The first two washes included 30-minute incubations in 4HT free media. Cells were then seeded with or without 4HT for 48 hours before analysis.

### Primary B cell isolation and culture

RosetteSep and EasySep negative isolation kits (Stemcell Technologies) were used sequentially to isolate CD19 + B cells by negative selection, with the following modifications made to the manufacturer's protocols. For RosetteSep, 40 μL antibody mixture was added per mL of blood and before Lymphoprep density medium was underlayed, prior to centrifugation. For EasySep, 10 μL antibody mixture was added per mL of B cells, followed by 15 μL magnetic bead suspension per mL of B cells. After negative selection, the cells were washed twice with 1x PBS, counted, and seeded for EBV infection studies. Cells were cultured in RPMI-1640 (Invitrogen) supplemented with 10% FBS and penicillin-streptomycin in a humidified chamber at 37 °C and 5% CO2. Cells were cultured in RPMI-1640 supplemented with 10% FBS and penicillin-streptomycin in a humidified incubator at 37 °C and at 5% CO2.

### Antibodies and reagents

Antibodies against the following proteins were used in this study: IMPDH1 (Cell Signaling Technology, #57068), IMPDH2 (Cell Signaling Technology, # 35914S), GAPDH (EMD Millipore, MAB374), LMP1 (Abcam, ab78113), LMP2A (Abcam, ab59028), EBNA2 PE2 (a gift from Fred Wang), DDX1 (Bethyl, A300-521A-M), Myc (Santa Cruz Biotechnology, SC-40), p100/p52 (EMD Millipore, 05-361), TRAF1 (Cell Signaling Biotechnology, #4715S), HA tag antibody (Cell Signaling Technology, # 3724), Fas-APC (Biolegend, 305612), ICAM-1-PE (BD Bioscience, 555511), Caspase 3 (Cell Signaling Technology, #9662), EBNA1 (a gift from Jaap Middeldorp). The following reagents were utilized in this study at the indicated concentration unless otherwise noted: DMSO (Fisher, BPBP231-100), mycophenolic acid (Selleckchem Cat#S2487, 1 μM), doxycycline hyclate (Sigma, D9891-1G, 250 ng/mL), GTP (Roche, 10106399001, 100 μM), Pifithrin-α (MedChemExpress, HY-15484, 10 μM), pan-PKCi staurosporine (Selleckchem, S1421, 100 nM [105]), JNK inhibitor SP 600125 (Apex Bio, A4604, 10 μM [106]), ERK inhibitor SCH772984 (Selleckchem, S7101, 10 μM [107]), p38 inhibitor Adezmapimod

(Selleckchem, S1076, 10 µM [108]), and IKKβ/IKK-2 inhibitor VIII (Apex Bio, A3485, 1 µM [48]), Etoposide (Sigma Aldrich, E1383-25MG, 50 µM [109]).

## B95.8 EBV preparation and B-cell infection

B95-8 EBV stocks were prepared from B95-8 producer cells as previously described [110,111] and stored at −80 °C. Infectious titer of freshly thawed EBV was determined by primary B-cell transformation assay. Freshly isolated, de-identified, discarded CD19 + peripheral blood B cells were seeded in RPMI1640 with 10% FBS at a concentration of one million cells/mL for infection studies at an EBV multiplicity of infection of 0.1.

## Metabolite extraction and metabolomic analysis

Metabolites were extracted according to Asara lab published protocols [112]. Twenty-four hours after doxycycline (250 ng/mL, Sigma Cat#D9891-1G) addition, uninduced and LMP1-induced Burkitt cells were pelleted and resuspended in fresh RPMI/FBS. After two hours, metabolites were extracted from two million cells using 4 mL of −80 °C 80% methanol, made from HPLC grade water (Sigma, 270733-1L) and LC-MS-grade methanol (Fisher, A456-1). Cells were suspended in 80% methanol by vortexing and pipetting, and extraction was performed overnight in a 4 degree C room on dry ice to ensure metabolite stability. Cell debris was pelleted in a 15 mL conical at 14,000 g for 5 minutes at 4 degrees C and supernatant was collected on dry ice. An additional round of extraction was performed using 0.5 mL of fresh 80% methanol (−80 °C). Cell debris were vigorously suspended using a combination of intense vortexing and pipetting with a p1000. A total of 4.5 mL of extracted metabolites were split into three 1.5 ml Eppendorf tubes and then dried using a Speedvac for ~6–8 hours. Dried pellets were stored at −80 until resuspension in HPLC water and LC-MS/MS analysis. LC-MS/MS was performed as published [112]. Peak area integrated TIC were utilized for relative comparisons of metabolites between samples.

## Targeted mass spectrometry

Samples were re-suspended using 20 µL HPLC grade water for mass spectrometry. 5–7 µL were injected and analyzed using a hybrid 6500 QTRAP triple quadrupole mass spectrometer (AB/SCIEX) coupled to a Prominence UFLC HPLC system (Shimadzu) via selected reaction monitoring (SRM) of a total of 298 endogenous water soluble metabolites for steady-state analyses of samples. Some metabolites were targeted in both positive and negative ion mode for a total of 309 SRM transitions using positive/negative ion polarity switching. ESI voltage was + 4950V in positive ion mode and −4500V in negative ion mode. The dwell time was 3 ms per SRM transition and the total cycle time was 1.55 seconds. Approximately 9–12 data points were acquired per detected metabolite. Samples were delivered to the mass spectrometer via hydrophilic interaction chromatography (HILIC) using a 4.6 mm i.d x 10 cm Amide XBridge column (Waters) at 400 µL/min. Gradients were run starting from 85% buffer B (HPLC grade acetonitrile) to 42% B from 0-5 minutes; 42% B to 0% B from 5–16 minutes; 0% B was held from 16-24 minutes; 0% B to 85% B from 24-25 minutes; 85% B was held for 7 minutes to re-equilibrate the column. Buffer A was comprised of 20 mM ammonium hydroxide/20 mM ammonium acetate (pH = 9.0) in 95:5 water:acetonitrile. Peak areas from the total ion current for each metabolite SRM transition were integrated using MultiQuant v3.0 software (AB/SCIEX).

## CRISPR/Cas9 mutagenesis

B-cell lines with stable Cas9 expression were established as described previously [113]. sgRNA constructs were generated as previously described [114] using sgRNA sequences from the Broad Institute Avana or Brunello libraries. CRISPR editing was performed as previously described [115]. Briefly, lentiviruses encoding sgRNAs were generated by transient transfection of 293T cells with packaging plasmids pasPAX2 (Addgene, Plasmid #12260) and pCMV-VSV-G (Addgene,

Plasmid #8454) and pLentiGuide-Puro (Addgene, Plasmid #52963) plasmids. P3HR-1, Daudi, MUTU I, GM12878, and GM13111 cells stably expressing Cas9 were transduced with the lentiviruses and selected with 3 µg/mL puromycin (Thermo Fisher, Cat#A1113803) for three days before replacement with antibiotic-free media. CRISPR editing was confirmed by immunoblotting 3 days post puromycin selection. For IMPDH1/IMPDH2 double knockout, in addition to the pLentiGuide Puro vector harboring IMPDH1 sgRNA a, second transduction was performed 3 days after the addition of puromycin using pLenti SpBsmBI sgRNA Hygro vector (Addgene, Plasmid # 62205) harboring the IMPDH2 sgRNA. Hygromycin (100 µg/mL, Thermo Fisher, Cat #10687010) selection was carried out two days post the second transduction. The sgRNAs used in this study were constructed using oligos based on the sequences below:

## ChIP and ChIP-qPCR

Five million cells were pelleted, washed using 1x PBS, and then fixed with 1% formaldehyde (Sigma, Cat#252549-100ML) in RPMI1640 for 20 minutes at room temperature. Cross-linking was quenched by adding 125mM glycine at room temperature for 5 minutes. After two more 1x PBS washes, Cells were lysed in 1 mL of ChIP lysis buffer (50 mM Tris, 10 mM EDTA, 1% SDS) supplemented with 1x cOmplete, EDTA-free Protease Inhibitor Cocktail (Pierce). Lysates were divided up into 250 µL aliquots and then sonicated with Bioruptor (Diagenode) with 30s on, 30s off for 12 cycles. Shearing of chromatin was confirmed by electrophoresis through a 0.8% Agarose gel. Sonicated chromatin was diluted 1:10 with ChIP dilution buffer (1.2 mM EDTA, 16.7 mM Tris, 167 mM NaCl, 0.01% SDS, 1.1% Triton X-100) and incubated with antibodies of interest or control IgG (Cell Signaling Technology, #2729S) antibodies overnight at 4c. Antibodies used for ChIP were anti-H3K9me3 (Active Motif, 39062), anti-H3K9Ac (Cell Signaling Technology, 9649S) anti-H3K27ac (Abcam, ab4729), and anti-H3K27me3 (Active Motif, 39155). Protein-DNA complexes were precipitated with protein A/G magnetic beads added at the same time as the antibodies (Pierce, 88803). Magnetic beads were washed extensively (washed twice with a lower salt buffer (150 mM NaCl, 2 mM EDTA, 20 mM Tris, 0.1% SDS, 1% Triton X-100) and then a high-salt buffer (500 mM NaCl, 2 mM EDTA, 20 mM Tris, 0.1% SDS, 1% Triton X-100), and once with LiCl buffer (0.25 M LiCl, 1% NP-40, 1% sodium deoxycholate, 1 mM EDTA, 10 mM Tris) and finally TE buffer (10 mM Tris, 1 mM EDTA). Each wash was 1 mL volume. Buffers were removed by placing solutions on magnet for 1 minute. Chromatin was eluted in Elution buffer (100 mM NaHCO3, 1% SDS) and reverse cross-linked at 65 °C for 2 hours. QIAquick PCR purification kits were used to purify the immunoprecipitated DNA, followed by qPCR with PowerUp SYBR green PCR master mix on a CFX Connect Real-Time PCR Detection System (Bio-Rad). qPCR was performed using 1 µL of eluted DNA per reaction, 0.5 µM FP and 0.5 µM RP for a total volume of 10 µL per reaction. Delta Ct values normalized to the percentage of input DNA. ChIP-PCR was performed using the primers below:

## Extracellular EBV genome copy number analysis

Treated cells were harvested 96 hours after treatment spun at 2000 g x 5 min to pellet cells. Supernatants were syringe filtered through a 0.45 µm filter. Afterwards, 500 µL of supernatant was collected and first treated with 20 µL DNase I (Thermo Fisher, 90083) for 1 hour at 37c. Afterwards, supernatants were supplemented with 100 µL 10% SDS and 30 µL proteinase K (New England Biolabs, P8107S) treatment for 1 hour at 65c. After DNAse and proteinase K treatment, an equal volume of phenol-chloroform-IAA was added, the sample was mixed vigorously via vortexing, and the supernatant was placed on ice for 10 minutes. Samples were spun at 4c for 45 minutes at 17,000 g. Aqueous layer was removed and supplemented with 1/10th volume of 3M sodium acetate, 2 volumes of ice-cold absolute ethanol, and 10 µL of glycogen (to aid with visualization of precipitated pellet) (Thermo Fisher, R0561). Precipitation was carried out overnight in a -20c freezer. Precipitated DNA was spun at 4c at 17,000 g for 1 hour and pelleted glycogen/DNA was washed 2x with room temperature 70% ethanol (with 17,000 g spins for 1 hour between). Ethanol was removed via aspiration and pellets were allowed to air-dry until nearly dried before dissolving in 20 µL of buffer AE (Qiagen, 19077). To quantitate viral genome

**Table 3. BALF5 qPCR primer sequences.**

| Primer | Sequence |
|---|---|
| BALF5-FP | GAG CGA TCT TGG CAA TCT CT |
| BALF5-RP | TGG TCA TGG ATC TGC TAA ACC |

copies, qPCR was performed using PowerUp SYBR green PCR master mix on a CFX Connect Real-Time PCR Detection System (Bio-Rad). Eluted extracellular DNA was compared to pHAGE-BALF5 serial dilution standard curve to assess limit of detection (Table 3). The following primers were used to conduct BALF5 qPCR:

### Flow cytometry

For Fas/ICAM-1 staining, cells were harvested and washed twice with 1x PBS supplemented with 2% FBS (Gibco). Cells were then incubated 45 minutes at room temperature with indicated antibodies at manufacturer concentrations. Cells were washed twice more after incubation to remove excess antibody and immediately analyzed on a flow cytometer. For 7-AAD (Thermo Fisher, Cat#A1310) viability assays, cells were harvested and washed twice with 1x PBS supplemented with 2% FBS (Gibco). Cells were then incubated with a 1 µg/mL 7-AAD solution in 1x PBS/2% FBS for five minutes at room temperature, protected from light. Cells were then analyzed via flow cytometry. For Annexin V (Biolegend, Cat#640906) vs 7-AAD assays, cells were harvested and incubated with a solution containing: 2.5 µL of Annexin V FITC antibody, 2.5 µL of a 20 ng/mL solution of 7-AAD, and 100 uL of with 1x PBS supplemented with 2% FBS per sample. Cells were incubated in this Annexin V/7-AAD solution for 20 minutes at room temperature, protected from light, before immediate analysis. For CFSE labeling and cell proliferation assays, CellTrace CFSE (Invitrogen, Cat#C34554) solution was prepared according to manufacturer's instructions. 10^7 cells were resuspended and incubated in one mL of CellTrace working solution for ten minutes in a 37°C/5% CO2 incubator, protected from light, with the cap of the vessel ajar. Five mL of RPMI 1640 with 10% FBS was added to the stained cells protected from light. Cells were incubated at room temperature for five minutes, protected from light, to remove free dye and prevent toxicity. Cells were then pelleted by centrifugation (300g x 5 minutes, room temperature) and resuspended in fresh RPMI with 10% FBS three times before the cells were seeded in complete media for proliferation analysis experiments at a concentration of 300,000 cells/mL. Labeled cells were analyzed by flow cytometry using a BD FACSCalibur instrument and analysis was performed with FlowJo V10.

### EdU vs PI cell cycle analysis

First, cells were supplemented with 10 µM EdU and incubated at 37c/5% $CO_2$ for 30 minutes. Cells were collected and washed 2x with 1x PBS. Cells were then fixed and permeabilized using BD Fix/Perm kit (BD, Cat# 554714). After fixation and permeabilization, cells were resuspended in a solution consisting of 800 µL of 1x BD Perm Wash, 16 uL of RNAse A (Thermo Fisher, Cat# EN0531), and 16 µL of Propidium Iodide (Thermo Fisher, Cat# P3566). Cells were then incubate for 15 minutes at 37c. After PI staining, cells were washed 2x with 1x PBS supplemented with 1% BSA before click chemistry was performed. Cells were resuspended in a click chemistry buffer consisting of 2 mM Copper Sulfate Pentahydrate and 10 mM Sodium ascorbate in 1x PBS; 100x Copper Sulfate Pentahydrate and Sodium ascorbate stock solutions were prepared fresh each time. Additionally, 10 µM Alexa Fluor 488 Azide (Thermo Fisher Cat# A10266) was added to the click chemistry buffer immediately before resuspending the cells. Cells were incubated in 1 mL of this 488 Azide/Click Chemistry buffer for 1 hour at 37c. After click chemistry steps were performed, cells were washed 2x with 1x PBS before performing flow cytometry. EdU-488 was analyzed logarithmically while PI levels were analyzed linearly. Labeled cells were analyzed by flow cytometry using a BD FACSCalibur instrument and analysis was performed with FlowJo V10.

## Immunofluorescence assays

For Normal Oral Keratinocyte (NOK) staining, 20,000 cells were seeded into a chamber slide well per condition (Ibidi, 81201). For B cells, 50,000 cells were seeded on to a chamber slide well per condition (Fisher, NC9811708). B cells were dried on to slides for 30 minutes at 37c. All cell conditions were fixed for 10 minutes at room temperature using 4% Paraformaldehyde in 1x PBS – 125 µL for NOK chambers and 20 µL for B cell chamber slides. After fixation, slides were washed 2x using 1x PBS; 200 µL of 1x PBS was used per wash per chamber for NOK cells while B cell mounted slides were briefly washed using 1x PBS rinsed over the mounted wells. Cells were permeabilized using 0.5% triton in 1x PBS (125 µL for NOK chambers and 20 µL for B cell chamber slides) at room temperature for 5 minutes. Cells were then blocked for one hour at room temperature using a 1% Low-IgG BSA solution in 1x PBS (MPBio, 219989725). After blocking, primary antibodies were utilized at a 1:200 dilution in the same blocking solution overnight at 4c in a humid box (125 µL for NOK chambers and 20 µL for B cell chamber slides). Primary antibody was removed by washing 2x with 1x PBS as previously before adding 1:500 dilutions of secondary antibody for one hour at 37c (125 µL for NOK chambers and 10 µL for B cell chamber slides). Secondary antibody was removed by washing 2x with 1x PBS before 1:10,000 Hoechst 33258 stain in 1x PBS was added to visualize nuclei (Thermo Fisher, 62249). Hoechst staining was performed at 37c for 15 minutes. Finally, slides were dried fully using a combination of aspiration and Kim wipes to dry wells before single drops of ProLong Gold anti-fade mountant were added per well and slides were placed (Thermo Fisher, P36970). Slides were allowed to cure for at least 24 hours at room temperature before imaging. Image acquisition and analysis was performed with Zeiss LSM 800 instrument and with Zeiss Zen Lite. Further Z-stack analysis was performed using ImageJ FIJI package. Primary antibodies used for immunofluorescence included: Rabbit anti IMPDH1 (Proteintech, 22092-1-AP), Rabbit anti IMPDH2-Coralite488 (Proteintech, CL488-12948), Mouse anti IMPDH2 (Proteintech, 67663-1-Ig), Rabbit anti HA-tag (Cell Signaling Technology, 3724), Mouse anti HA-tag (Biolegend, 901514), mouse anti LMP1 (Abcam, ab78113). Secondary antibodies used include: Goat anti-mouse Alexa Fluor 488 (Invitrogen, A11029), Goat anti-Rabbit Alexa Fluor 488 (Invitrogen, A11008), Goat anti-mouse Alexa Fluor 568 (Invitrogen, A11031), and Goat anti-Rabbit Alexa Fluor 568 (Invitrogen, A11036).

## CellTiter-Glo

CellTiter-Glo viability assay (Promega) was performed according to the manufacturer's protocol at the indicated time points. A total of 50 µL cells in PBS were used per assay according to manufacturer instruction.

## Software/data presentation

Statistical analysis was assessed with Student's t test using GraphPad Prism 7 software, where NS = not significant, $p > 0.05$; * $p < 0.05$ ** $p < 0.01$; *** $p < 0.005$. And graphs were made using GraphPad Prism 7. Metabolic pathway analyses were performed using Metaboanalyst 6.0 platform [20].

## Supporting information

**S1 Table. Akata and daudi metabolomics.** Peak area intensities derived from both Akata and Daudi Burkitt lymphoma cells selectively expressing indicated LMP1 or LMP1 mutants and following fold change and statistical analyses. (XLSX)

**S2 Table. Shared significant metabolites in Akata vs Daudi.** Comparison of significantly ($p < 0.05$) changed metabolites found in Akata and Daudi Burkitt lymphoma cells selectively expressing LMP1. Relative fold change of metabolites from LMP1 + vs LMP1- cells are shown. (XLSX)

**S3 Table. Daudi LMP1 inhibitor metabolomics.** Peak area intensities derived from Daudi Burkitt lymphoma cells selectively expressing LMP1 and treated with indicated small molecule inhibitors: PKC inhibitor (PKCi) staurosporine (100nM), JNK inhibitor (JNKi) SP600125 (10 μM), ERK inhibitor (ERKi) SCH772984 (10 μM), p38 inhibitor (p38i) adezmapimod (10 μM) or IKKβ inhibitor (IKKβi) IKK-2 VII (1 μM).
(XLSX)

**S1 Fig. Validation of conditional LMP1 Burkitt cell models.** (A) Immunoblot analysis of whole cell lysates (WCL) from Daudi cells mock induced or induced for LMP1 expression by doxycycline (250ng/mL) for 24 hours. LMP1 expression induced non-canonical NF-κB activity, as judged by processing of the p100 precursor into the p52 subunit, and induced expression of the well characterized LMP1/NF-κB target TRAF1. (B) FACS analysis of plasma membrane Fas abundance in Daudi cells mock induced or induced for LMP1 expression for 24 hours, as in (A). Conditional LMP1 expression highly induced expression of the well characterized LMP1 target Fas. (C) Immunoblot analysis of WCL from Akata cells mock induced or induced for LMP1 expression as in (A), indicating LMP1 induction of non-canonical NF-κB activity and LMP1 target TRAF1 expression. (D) FACS analysis of plasma membrane ICAM-1 abundance in Akata cells mock induced or induced for LMP1 expression as in (A). ICAM-1 rather than Fas was analyzed as basal Fas expression is aberrantly elevated in Akata cells.
(TIF)

**S2 Fig. EBV latency state dependent MPA effects on proliferation and survival.** (A) FACS analysis of CFSE levels following 96 hours of treatment with the indicated MPA dosages in the indicated cell lines. (B) FACS analysis of 7-AAD uptake following 48 hours of treatment of indicated cell lines with the indicated MPA dosages. (C) FACS analysis of MUTU I versus III 7-AAD uptake following 48 hours of treatment with the indicated MPA dosages. (D) FACS analysis of P3HR-1 versus Jijoye cell 7-AAD uptake following 48 hours of treatment of MUTUI vs III with the indicated MPA dosages. (E) Immunoblot analysis of WCL from MUTU I, III or GM12878 treated with DMSO or 1 μM MPA with or without 100 μM GTP rescue for 48 hours. Cells treated with 50 μM etoposide as a control for ATM and ATR phosphorylation are included. Shown are the normalized phospho-ATM:ATM and phospho-ATR:ATR ratios calculated by densitometry analysis, with values in DMSO-treated cells set to 1. N.D., not determined.
(TIF)

**S3 Fig. FACS analysis of MPA effects on GTP-dependent EBV+B-cell viability in latency I versus III.** Shown are representative FACS plots from n = 3 replicates of the indicated cell lines treated with DMSO, 1 μM MPA or 1 μM MPA + 100 μM GTP for 48 hours, as shown in Fig 2F.
(TIF)

**S4 Fig. p53 inhibition does not rescue MPA-induced LCL death.** (A) Representative FACS plots from n = 3 replicates of GM15892 or GM12878 cells treated with DMSO vehicle, 1 μM MPA and/or 10 μM pifithrin-α for 48 hours, as indicated. (B and C) Mean ± SD percentages of double 7-AAD+/Annexin V+ GM12878 (B) or GM15892 (C) treated with DMSO, MPA and/or pifithrin-α as in (A).
(TIF)

**S5 Fig. P53 knockout does not rescue MPA-induced LCL death.** (A) Immunoblot analysis of CRISPR p53 editing in GM12878 and GM15892 LCLs. Shown are representative immunoblots from n = 3 replicates of WCL from LCLs expressing the indicated control or p53-targeting sgRNA. (B and C) Representative FACS plots of 7-AAD+/Annexin V+ stained GM12878 (B) or GM15892 cells (C) transduced with control or p53 targeting single guide RNAs (sgRNA) from n = 3 replicates. Cells transduced with lentiviruses expressing the indicated sgRNAs, puromycin selected for four days and then treated with DMSO or MPA, as indicated for 48 hours.
(TIF)

**S6 Fig. P53 knockout does not rescue MPA-induced LCL death.** (A and B) Mean±SD 7-AAD+/Annexin V+ from n=3 replicates of Cas9+GM12878 (A) or GM15892 (B) expressing the indicated control or p53 targeting sgRNA, as in S5 Fig. FACS 7-AAD+/Annexin V+ analyses were performed after 48 hours of treatment.
(TIF)

**S7 Fig. Caspase or RIP1 inhibition do not rescue LCLs from MPA-induced death.** (A) Caspase 3/7 activity assays. Shown are mean±SD Caspase 3/7 Glo values from n=3 replicates of GM12878 and GM15892 LCLs treated with DMSO, MPA (1 μM) and/or the pan-caspase inhibitor ZVAD-FMK (40 μM) for 48 hours, as indicated. (B and C) GM12878 (B) or GM15892 (C) LCLs were treated with DMSO vehicle or with MPA (1 μM), the pan-caspase inhibitor ZVAD-FMK (40 μM), and/or the necroptosis pathway RIP1 kinase inhibitor Necrostatin-1s (20 μM) as indicated. FACS analysis of 7-AAD uptake and annexin V positivity was performed 48 hours later. Representative FACS plots from n=3 independent replicates are shown.
(TIF)

**S8 Fig. Caspase or RIP1 inhibition do not rescue LCLs from MPA-induced death.** (A and B) GM12878 (A) and GM15892 (B) LCLs were treated with DMSO, MPA (1 μM), ZVAD-FMK (40 μM), and/or Necrostatin-1s (20 μM), as in S7 Fig. Shown are mean±7-AAD uptake values from n=3 replicates.
(TIF)

**S9 Fig. Genetic perturbation of IMPDHs impact growth and viability of lymphoblastoid cell lines.** (A) Immunoblot analysis of IMPDH1 or IMPDH2 depletion. WCL from Cas9+MUTU I or GM12878 expressing were subjected to immunoblot analysis at the indicated sgRNA at two days post-puromycin selection of successfully transduced cells. *=non-specific band. (B) Immunoblot analysis of IMPDH1 or IMPDH2 depletion in Daudi versus GM13111 WCL, as in (A). (C) Immunoblot analysis of IMPDH1 or IMPDH2 depletion in P3HR-1 versus GM12878 WCL, as in (A). (D) FACS analysis of 7-AAD uptake in Cas9+P3HR-1 expressing the indicated sgRNA at 8 days post-puromycin selection of cells transduced by sgRNA expressing lentivirus. (E) FACS analysis of 7-AAD uptake in Cas9+GM12878, as in (D). FACS plots and immunoblots are representative of n=3 independent replicates.
(TIF)

**S10 Fig. EBV latency gene effects on IMPDH1 and IMPDH2 expression.** (A) Normalized *IMPDH1* (left) versus *IMPDH2* (right) reads from RNAseq analysis of primary human B cells at the indicated days post-infection by the EBV B95.8 strain. Shown are mean±SD values from n=3 replicates [42,63–65]. (B) Relative IMPDH1 (left) versus IMPDH2 (right) protein abundances from tandem mass tagged multiplexed mass spectrometry proteomic analysis of primary human B-cells at the indicated days post-infection by the EBV B95.8 strain. Shown are mean±SD values from n=4 replicates [42]. (C) Immunoblot analysis of WCL from primary human B-cells infected by B95.8 EBV at the indicated days post infection (DPI). DDX1 was used as a load control as its levels remain relatively unchanged between uninfected versus infected primary B-cells [42]. (D) Immunoblot analysis of WCL from 2-2-3 EBNA2-HT LCLs, which harbor a conditional EBNA2 allele that is fused to a modified estrogen receptor ligand binding domain, whose activity requires the presence of 4-hydroxytamoxifen (4-HT). WCL were generated at 48 hours post EBNA2 inactivation by 4-HT washout. (E) Immunoblot analysis of WCL from Daudi cells mock induced or induced for wildtype, TES1 point mutant (TES1m), TES2 point mutant (TES2m), or double TES1m/TES2m LMP1 for 24 hours by 250 ng/mL doxycycline. Immunoblots are representative of n=3 independent replicates.
(TIF)

**S11 Fig. MPA induces IMPDH cytoophidium formation in normal oral keratinocytes.** Representative images of normal oral keratinocytes treated with DMSO or with MPA (10 μM) for 24 hours and then stained for IMPDH1, IMPDH2 or with the Hoechst 33258 nuclear dye. Composite Z-stack images are also shown. Cells stained with antibody isotype controls are shown at bottom. Images are representative on n=3 replicates. White scalebar indicates 10 μM distance.
(TIF)

**S12 Fig. LMP1 does not increase Burkitt cell cytoophidium levels.** (A and B) Representative images from n = 3 replicates of Daudi cells mock-induced or induced for LMP1 expression for 24 hours or treated with MPA (1 μM) for 24 hours, stained and analyzed by confocal microscopy. (A) Cells were stained with αIMPDH1 and αHA-LMP1. (B) Cells were stained with αIMPDH2 and αHA-LMP1. White arrows indicate cytoophidium. White scalebar indicates 10 μM distance. (TIF)

**S13 Fig. Cytoophidium are not observed in LCLs.** Representative images of GM12878 stained by Hoechst, anti-IMPDH1 or IMPDH2 and analyzed by confocal microscopy. White scalebar indicates 10 μM distance. (TIF)

**S14 Fig. LMP1 expression sensitizes Daudi cells to MPA-driven death in a partially GTP dependent manner.** Representative FACS plots from n = 3 replicates of Daudi cells mock induced or induced for WT, TES1m, TES2m or TES1m+TES2m LMP1 expression for 24 hours and then treated with 1 μM MPA ± 100 μM GTP for 96 hours as indicated and as in Fig 4. Shown are FACS analysis of 7-AAD uptake versus Annexin V positivity. (TIF)

**S15 Fig. LMP1 TES1 and TES2 signaling sensitize EBV-negative BL41 Burkitt cells to MPA-driven apoptosis in a partially GTP dependent manner.** Representative FACS plots from n = 3 replicates of BL41 cells mock induced or induced for WT, TES1m, TES2m or TES1m+TES2m LMP1 expression for 24 hours and then treated with 1 μM MPA ± 100 μM GTP for 96 hours as indicated. Shown are FACS analysis of 7-AAD uptake versus Annexin V positivity. (TIF)

**S16 Fig. LMP1 TES1 and TES2 signaling sensitize EBV-negative BL41 Burkitt cells to MPA-driven apoptosis in a partially GTP dependent manner.** Mean ± SD percentages from n = 3 replicates from n = 3 replicates as in S15 Fig of BL41 cells mock induced or induced for the indicated LMP1 construct for 24 hours and then treated with DMSO or 1 μM MPA ± 100 μM GTP for 96 hours. *P < 0.05, **P < 0.01, ***P < 0.005. (TIF)

**S17 Fig. Effects of LMP1 TES1 versus TES2 signaling on Akata Burkitt metabolome remodeling.** (A) Volcano plot of LC-MS metabolomic analysis of n = 6 replicates of EBV-negative Akata cells mock induced or doxycycline induced for LMP1 TES1m expression for 24 hours. Metabolites with higher abundance in LMP1 TES1+ cells have positive fold change values, whereas those higher in mock induced cells have negative fold change values. Selected metabolites are highlighted by red circles and annotated. (B) Volcano plot of LC-MS metabolomic analysis of n = 6 replicates of EBV-negative Akata cells mock induced or doxycycline induced for LMP1 TES2m expression for 24 hours, with selected metabolites highlighted as in (A). (C) Volcano plot of LC-MS metabolomic analysis of n = 6 replicates of EBV-negative Akata cells doxycycline induced for TES1m vs WT LMP1 expression for 24 hours, with selected metabolites highlighted. Replicates for this cross-comparison were induced side by side, prepared for and analyzed by LC-MS together on the same day to minimize batch effects. (D) Volcano plot of LC-MS metabolomic analysis of n = 6 replicates of EBV-negative Akata cells doxycycline induced for TES2m vs WT LMP1 expression for 24 hours, with selected metabolites highlighted. Replicates for this cross-comparison were induced side by side, prepared for and analyzed by LC-MS together on the same day to minimize batch effects. (TIF)

**S18 Fig. MPA does not de-repress EBNA2 or LMP2A protein expression in Burkitt cells.** (A) Immunoblot analysis of WCL from Daudi (left) versus MUTU I (right) treated with DMSO, 1 μM MPA ± 100 μM GTP for 24 hours. WCL for latency III GM12878 LCLs was run in the rightmost lane as a positive control for EBNA2 and LMP2A expression. Daudi contain an EBV genomic deletion that knocks out the EBNA2 gene. (B) ChIP-qPCR analysis of Daudi cells treated with DMSO or 1 μM MPA for 72 hours, using the indicated ChIP antibodies and qPCR primers specific for the LMP1p region. (C) FACS

analysis of MPA effects on Burkitt cell cycle. Shown are representative FACS plots of 5-ethynyl-2′-deoxyuridine (EdU) vs propidium iodide (PI) levels in MUTU I (top) versus Rael Burkitt cells treated with DMSO or MPA (1 µM) for 48 hours. Gating was constructed using DMSO treated Burkitt cells. Percentages of total cells in each cell cycle phase are shown. Data is representative of n = 3 independent experiments. (D) Analysis of MPA effects on secreted EBV copy number. Shown are the mean ± S.D. values from qPCR analysis using primers shown in Table 2 of DNAse-treated supernatants from Rael or MUTU I cells, which were incubated with DMSO, MPA, GTP and/or the HDAC inhibitor sodium butyrate (NaB) for 96 hours. Shown to the right is the standard curve generated from the indicated amounts of the pHAGE-BALF5 plasmid. The limit of detection was determined by the pHAGE-the BALF5 serial dilution curve. Statistical analysis of Rael and Mutu I supernatants were performed by cross-comparison with the respective DMSO-treated qPCR Ct result. n.s. = not significant. (TIF)

## Acknowledgments

We thank Jaap Middeldorp providing the EBNA1 monoclonal antibody.

## Author contributions

**Conceptualization:** Eric M. Burton, Shaowen White, Benjamin E. Gewurz.

**Data curation:** Eric M. Burton, Davide Maestri, Jin-Hua Liang, John M. Asara.

**Formal analysis:** Eric M. Burton, Davide Maestri, Shaowen White, John M. Asara.

**Funding acquisition:** Eric M. Burton, Benjamin E. Gewurz.

**Investigation:** Eric M. Burton, Shaowen White, Jin-Hua Liang, Bidisha Mitra, John M. Asara.

**Methodology:** Eric M. Burton, Shaowen White, Bidisha Mitra, John M. Asara.

**Project administration:** Benjamin E. Gewurz.

**Resources:** John M. Asara.

**Supervision:** Benjamin E. Gewurz.

**Validation:** Eric M. Burton, Jin-Hua Liang, John M. Asara.

**Visualization:** Eric M. Burton, Davide Maestri.

**Writing – original draft:** Eric M. Burton, Benjamin E. Gewurz.

**Writing – review & editing:** Eric M. Burton, Benjamin E. Gewurz.

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
