## [Decision Letter · Decision Letter 0]

11 Dec 2024

PPATHOGENS-D-24-02482

Epstein-Barr Virus Latent Membrane Protein 1 Subverts IMPDH pathways to drive B-cell oncometabolism

PLOS Pathogens

Dear Dr. Gewurz,

Thank you for submitting your manuscript to PLOS Pathogens. After careful consideration, we feel that it has merit but does not fully meet PLOS Pathogens's publication criteria as it currently stands. Therefore, we invite you to submit a revised version of the manuscript that addresses the points raised during the review process.

Please submit your revised manuscript within 30 days Feb 09 2025 11:59PM. If you will need more time than this to complete your revisions, please reply to this message or contact the journal office at plospathogens@plos.org. Please include the following items when submitting your revised manuscript:

We look forward to receiving your revised manuscript.

Kind regards,

Nancy Raab-Traub

Academic Editor

PLOS Pathogens

Blossom Damania

Section Editor

PLOS Pathogens

Sumita Bhaduri-McIntosh

Editor-in-Chief

PLOS Pathogens

orcid.org/0000-0003-2946-9497

Michael Malim

Editor-in-Chief

PLOS Pathogens

orcid.org/0000-0002-7699-2064

**Additional Editor Comments:**

The three reviews all appreciate the importance of the finding and the quality of the experimental data. However, the inability to pinpoint a mechanism reduces enthusiasm and all request additional experiments evaluating mechanism. I think it is likely that you may already have most of this data or can readily obtain it. Reviewer #1 wants to know how established LMP1 induced pathways, NF-kB, Erk, PKC, contribute to this and the contribution of p53 to the induced apoptosis. This is interesting given that the BL cell lines likely have mutant p53. Reviewer #2 is also interested in this and requests assessment of the MPA induced cell cycle arrest. This reviewer has other interesting questions and suggestions although assessment of viral replication is of tangential interest. Reviewer #3 summarizes nicely that "The only significant limitation of this study is that they stop short of defining exactly how LMP1 is upregulating IMPDH2 activity" and would like inclusion of any analyses of cytoophidium.

**Journal Requirements:**

https://journals.plos.org/plospathogens/s/submission-guidelines#loc-parts-of-a-submission

- TM on Lines: 579, and 581.

5) We have noticed that you have cited Tables 1 and 2 in the manuscript file but there is no corresponding table in the manuscript.  Please amend your manuscript to include this table noting that tables should not be uploaded as individual files.

6) We notice that your supplementary figures are uploaded with the file type 'Figure'. Please separate the supplementary figures from the main figure. Please amend the file type to 'Supporting Information'. Please ensure that each Supporting figure or table file has a legend listed in the manuscript after the references list.

7) Some material included in your submission may be copyrighted. According to PLOSu2019s copyright policy, authors who use figures or other material (e.g., graphics, clipart, maps) from another author or copyright holder must demonstrate or obtain permission to publish this material under the Creative Commons Attribution 4.0 International (CC BY 4.0) License used by PLOS journals. Please closely review the details of PLOSu2019s copyright requirements here: PLOS Licenses and Copyright. If you need to request permissions from a copyright holder, you may use PLOS's Copyright Content Permission form.

Potential Copyright Issues:

- Figures 1, and 7; Please confirm whether you drew the images / clip-art within the figure panels by hand. If you did not draw the images, please provide a link to the source of the images or icons and their license / terms of use; or written permission from the copyright holder to publish the images or icons under our CC BY 4.0 license. Alternatively, you may replace the images with open source alternatives. See these open source resources you may use to replace images / clip-art:

8) Please ensure that the funders and grant numbers match between the Financial Disclosure field and the Funding Information tab in your submission form. Note that the funders must be provided in the same order in both places as well.

**Reviewers' Comments:**

Reviewer's Responses to Questions

**Part I - Summary**

Reviewer #1: Burton et al have investigated the metabolic reprogramming of B lymphocytes by the Epstein-Barr virus (EBV) oncoprotein LMP1. The study is based on metabolomic analysis of Burkitt’s lymphoma (BL) cell lines with inducible LMP1 expression. The authors focused their analysis on the ability of LMP1 to increase the levels of xanthosine-5-phosphate (XMP), a precursor of guanine nucleotide biosynthesis. Interestingly, LMP1 expression was associated with an increased dependency on the activity of inosine monophosphate dehydrogenases (IMPDH), which catalyze the conversion of inosine monophosphate (IMP) to XMP, the first committed step in guanine nucleotide biosynthesis. More specifically, inhibition of IMPDH by mycophenolic acid (MPA) caused growth arrest of LMP1-negative BL cell lines and apoptosis of LMP1-expressing BL and EBV-transformed lymphoblastoid cell lines (LCL). The dependency of LCL survival on IMPDH was verified by genetic inactivation of the corresponding genes and showed a greater dependency on IMPDH2. Both carboxyl-terminal domains of LMP1 that play a critical role in B cell transformation were involved in XMP increased levels.

This study identifies important elements of B cell metabolic reprogramming that are caused by LMP1 in addition to relevant vulnerabilities that could be exploited for therapeutic purposes. However, it falls short of clarifying important mechanistic aspects of LMP1-mediated upregulation of XMP. LMP1 does not increase the levels of IMPDH1 or IMPDH2 and the authors speculate that LMP1 induces their activity. Nevertheless, they do not provide any direct evidence for LMP1-mediated activation of IMPDH1/2 or a relevant mechanistic explanation. It is known that actively proliferating cells depend on the activity of IMPDH and particularly IMPDH2, which is confirmed in the present study. In the absence of mechanistic data that link LMP1 to IMPDH1/2 activity the report by Burton et al offers limited advancement to the current state of the art.

Reviewer #2: Burton et al. argue that the Epstein-Barr virus (EBV) latency protein LMP1 both increases activity of inosine monophosphate dehydrogenase (IMPDH) and establishes a dependency of EBV-infected B cells on IMPDH for de novo purine synthesis. The authors begin by inducing LMP1 expression in Burkitt lymphoma Daudi (EBV+) and Akata (EBV-) lines, then analyzing changes in metabolite abundance. They reveal LMP1-dependent increases in purine biosynthesis intermediates (notably xanthosine-5-phosphate), and glutamine. Next, the authors showed a dose-dependent decrease in proliferation of Burkitt lines and lymphoblastoid cell lines (LCLs) after treatment with the IMPDH inhibitor MPA. Additionally, MPA treatment induced a dose-dependent increase in apoptosis in LCLs, which was rescued by GTP addition. Knockout of either IMPDH1 or IMPDH2 decreased growth and increased death of LCLs (with IMPDH2 knockout having a stronger phenotype), while knockout of both IMPDH1 and IMPDH2 was required to stunt growth of Burkitt lymphoma lines, highlighting a heightened dependence of LCLs on individual IMPDH isoforms. In order to test which of the LMP1 cytoplasmic domains were responsible for induction of GTP biosynthesis, the authors conditionally expressed mutant LMP1s in Burkitt lymphoma cells, with either the TES1 or TES2 domain mutated such that it is nonfunctional. Mutating either TES decreased the sensitivity of Burkitt lymphoma lines to IMPDH inhibition, with lines with dual TES mutants being mostly resistant to IMPDH inhibition. This finding, coupled with the finding that MPA was only able to induce Burkitt lymphoma cell death upon conditional LMP1 expression allow the authors to argue that LMP1 is necessary for MPA hypersensitivity. Next, the authors show that TES1 more strongly induces GTP metabolism compared to TES2, as Burkitt lines (Daudi) with TES2 mutants had a more significant increase in abundance of purine metabolites relative to uninduced Burkitt cells, compared to TES1 mutants. Last, the authors show that MPA treatment of Burkitt cell lines was sufficient to increase LMP1 expression, which was associated with a decrease in repressive histone modification and increase in activating histone modifications at LMP1 and EBV lytic gene promoters. Overall, the authors provide a potential explanation for why latency III LCLs are more sensitive to IMPDH perturbation than latency I Burkitt cells.

Reviewer #3: Proliferating B and T lymphocytes are singularly dependent on de novo purine biosynthesis [PMID: 415850] and many pharmacologic agents have been developed targeting this vulnerability. Among them is mycophenolic acid (MPA) an inhibitor of Inosine-5′-monophosphate dehydrogenase (IMPDH) which catalyzes the synthesis of xanthsoine-5-phosphate (XMP), the committed step in de novo guanosine production. MPA selectively impairs lymphocyte proliferation and is widely used as an anti-rejection agent in solid organ transplant recipients. This selectivity arises partly from use of salvage pathway by resting cells and from upregulation of the IMPDH2 gene in proliferating lymphocytes [PMID: 21873529, 1345808].

This manuscript by Burton and colleagues is a metabolomic study that uncovers a key role for the EBV LMP1 oncogene in regulating IMPDH activity. They begin by performing polar metabolite LC/MS profiling in the Daudi and Akata Burkitt lymphoma (BL) cell lines with and without LMP1 expression (Fig 1). They find that XMP levels are increased in both BLs and further show that LMP1 expression heightens sensitivity to MPA (Fig 2) and this effect is attributable to the combined effects of the CTAR1/TES1 and CTAR2/TES2 LMP1 domains (Fig 3). They conduct additional polar metabolite LC/MS profiling in the Daudi and Akata BL using LMP1 mutants to distinguish CTAR1/TES1 and CTAR2/TES2 domain effects. Notably, LMP1 does not alter IMPDH expression levels (Fig S4) and they suggest a likely mechanism: LMP1 promotes formation of IMPDH2 filaments which are known to increase its enzymatic activity [e.g., PMID: 31999252]. Finally, they make the interesting observation that LMP1 expression levels are increased by MPA treatment and provide evidence that this is due to lytic LMP1 expression.

This is a very well-done study that manages to provide interesting insights into effects of LMP1 on MPA sensitivity of B lymphocytes. Their findings suggest that MPA sensitivity is primarily a function of B cell activation rather than proliferation per se. This has obvious clinical implications and is impressive considering the extensive prior literature on the subject. (I’m not sure I share their belief that LMP1 is having effects markedly different than CD40L + BCR crosslinking as argued on lines 310-315 – see minor point below). The only significant limitation of this study is that they stop short of defining exactly how LMP1 is upregulating IMPDH2 activity. I agree that LMP1 induced formation of cytoophidium is a very likely mechanism to explain the observed increase in enzymatic activity despite unchanged IMPDH2 expression levels. Such IMPDH2 filaments can be demonstrated by either EM or IF [PMID: 30555474]. My one suggestion for further experiments is to perform IF (or EM if preferred) to fully evaluate this mechanism (the authors suggest they may have done this experiment already on lines 325-326 and it would be sufficient to include this data as a supplemental figure, provided there is an appropriate positive control for the assay’s ability to detect cytoophidium).

**Part II – Major Issues: Key Experiments Required for Acceptance**

Reviewer #1: The authors should determine post-translational changes of IMPDH1 and/or IMPDH2 that depend on LMP1 and activate the enzymes. Additional aspects that require further investigation, include the mechanism of MPA-induced apoptosis of LMP1-expressing cells and the role of LMP1-activated signaling pathways in XMP upregulation. For example, the authors should investigate the role of p53 in MPA-induced apoptosis of LMP1-expressing cells. In addition, it would be informative to determine the potential role of NF-kappaB and/or MAPK activation by LMP1 in XMP upregulation.

Reviewer #2: 1. BrdU/7-AAD experiment to test whether the MPA-mediated growth inhibition in Burkitt cell lines is at the G1/S phase where one might expect.

2. Figure S2E: show ATM/ATR phosphorylation in MPA-only treated LCLs. Add a positive control (e.g. lysate from cells treated with ionizing radiation).

3. Figure 4/5: test whether there are any growth differences between TES1 and TES2 mutants in the conditionally-expressing Burkitt cells to see whether the increased effect on purine metabolism by TES1 results in TES1 mutants growing slower.

4. Figure 6: measure extracellular EBV genome copy number in Burkitt cells after MPA treatment to see whether this lytic reactivation is productive.

Reviewer #3: 1) An experiment examining whether LMP1 induces formation of IMPDH2 filaments (cytoophidium) should be done (and, if results are negative, a positive control validating the assay is essential).

**Part III – Minor Issues: Editorial and Data Presentation Modifications**

Reviewer #1: Does doxycycline alone affect XMP levels?

In Figure 1E what does Try stand for?

Lines172-173: The sentence “Unexpectedly, MPA lowered …levels” should be extended as follows “Unexpectedly, MPA lowered ATM and ATR phosphorylation levels to a greater extent in latency III compared to latency I B cells”

Figure S4A and B: correct “IMDPH”

The text must be checked for typos i.e. line 128 xanthsoine, in line 152 the word “inhibition” is repeated twice.

The name initials in funding details are wrong.

Reviewer #2: 1. Figure 2B: missing asterisks for significance.

2. Figure 4E: the green color of bar plots is not consistent throughout the graph.

3. Figure 4E-F: the color key and +/- treatment indicators below the graphs do not correspond with each other. Green color says that both MPA and GTP are added, while the +/- indication says that only GTP is added.

4. Line 253: typo (“purine synthesis to support of GTP abundance”)

5. Line 272: typo (“neither LMP1 or lytic cycle”)

6. Line 334: typo (“CRIPSR”)

Reviewer #3: 1) Lines 22-24, 48-50, 80-81: I don’t share the view that LMP1 is sufficient to transform B cells. We know this is not true for human B cells [PMID: 38169736] and the murine studies cited to support that this is true for murine B cells do not exclude the possibility that somatic mutations are required to cooperate with LMP1 to induce transformation in that model.

2) Lines 69-79 - the EBERs, BARTs and BART-miRs need to be added to this.

3) lines 128-131 – “XMP . . . was the most highly LMP1 upregulated metabolite . . .” Based on what? GMP appears to be more upregulated in Daudi (Fig 1B), hypoxanthine more upregulated in Akata (Fig 1C).

4) Figure 1G is excellent, but please ensure that the same names are used as in Figs 1BCF. For example, I believe that XMP and Xanthosine-5-P refer to the same molecule. Please pick one name and stick to it throughout for all metabolites.

5) Why was Fas assayed in Daudi (Fig 1B) and ICAM-1 used for Akata (1D)? Use of the same assay or, better yet, showing both would be more convincing.

6) Since the signaling tail of LMP1 is on the C-terminus, it would be preferable if Fig7 were re-drawn such that the TMs are to the left of TES1/TES2 domains (I realize this is a pretty minor point)

7) In the results it is mentioned that Daudi LMP1 migrates at a different MW, but Fig 6B does not appear to show any convincing amount of LMP1 expression in these cells. Is LMP1 cropped out of the panel? If not, then there does not appear to be any credible induction of LMP1 in Daudi cells and the ChIPs in Fig 6C-F are uninformative and need to be done in Mutu I cells if any insights into LMP1 induction by MPA is to be gained.

8) A previous study examined donor-specific genetic determinants of MPA susceptibility among LCLs [PMID: 21396482] – if they think that any of these intersect with metabolism or LMP1 the authors should consider adding this to their discussion.

9) lines 310-315 – this may be true, but may also be a limitation of modeling GC reactions via 24 hour stimuli. The finding of IMPDH2 filaments in GC reactions [[PMID: 30555474] would argue its activity is markedly upregulated by physiologic lymphocyte activation.

10) lines 343-344 – one should not conclude that EBNA3C attenuates p53 function in latency III based on studies performed in osteosarcoma cells when the Luftig lab has clearly demonstrated that LCLs (an accepted latency III model) have intact p53 function despite EBNA3C expression.

PLOS authors have the option to publish the peer review history of their article (what does this mean? ). If published, this will include your full peer review and any attached files.

**Do you want your identity to be public for this peer review?** For information about this choice, including consent withdrawal, please see our Privacy Policy .

Reviewer #1: No

Reviewer #2: No

Reviewer #3: No

**Figure resubmission:**
---

## [Editor Report · Decision Letter 1]

1 Apr 2025

Dear Dr. Gewurz,

We are pleased to inform you that your manuscript 'Epstein-Barr Virus Latent Membrane Protein 1 Subverts IMPDH Pathways to Drive B-cell Oncometabolism' has been provisionally accepted for publication in PLOS Pathogens.

Best regards,

Nancy Raab-Traub, Ph.D.

Academic Editor

PLOS Pathogens

Blossom Damania

Section Editor

PLOS Pathogens

Sumita Bhaduri-McIntosh

Editor-in-Chief

PLOS Pathogens

orcid.org/0000-0003-2946-9497

Michael Malim

Editor-in-Chief

PLOS Pathogens

orcid.org/0000-0002-7699-2064

This is an elegant study with high impact. The authors have addressed all concerns and gone well beyond with inclusion of new data.
---

## [Editor Report · Acceptance letter]

Dear Dr. Gewurz,

We are delighted to inform you that your manuscript, "Epstein-Barr Virus Latent Membrane Protein 1 Subverts IMPDH Pathways to Drive B-cell Oncometabolism," has been formally accepted for publication in PLOS Pathogens.

Best regards,

Sumita Bhaduri-McIntosh

Editor-in-Chief

PLOS Pathogens

orcid.org/0000-0003-2946-9497

Michael Malim

Editor-in-Chief

PLOS Pathogens

orcid.org/0000-0002-7699-2064